# Subgame solving without common knowledge

**Brian Hu Zhang**
Computer Science Department
Carnegie Mellon University
bhzhang@cs.cmu.edu

**Tuomas Sandholm**
Computer Science Department, CMU
Strategic Machine, Inc.
Strategy Robot, Inc.
Optimized Markets, Inc.
sandholm@cs.cmu.edu

## Abstract

In imperfect-information games, subgame solving is significantly more challenging than in perfect-information games, but in the last few years, such techniques have been developed. They were the key ingredient to the milestone of superhuman play in no-limit Texas hold'em poker. Current subgame-solving techniques analyze the entire *common-knowledge closure* of the player's current information set, that is, the smallest set of nodes within which it is common knowledge that the current node lies. While this is acceptable in games like poker where the common-knowledge closure is relatively small, many practical games have more complex information structure, which renders the common-knowledge closure impractically large to enumerate or even reasonably approximate. We introduce an approach that overcomes this obstacle, by instead working with only low-order knowledge. Our approach allows an agent, upon arriving at an infoset, to basically prune any node that is no longer reachable, thereby massively reducing the game tree size relative to the common-knowledge subgame. We prove that, as is, our approach can increase exploitability compared to the blueprint strategy. However, we develop three avenues by which safety can be guaranteed. First, safety is guaranteed if the results of subgame solves are incorporated back into the blueprint. Second, we provide a method where safety is achieved by limiting the infosets at which subgame solving is performed. Third, we prove that our approach, when applied at every infoset reached during play, achieves a weaker notion of equilibrium, which we coin *affine equilibrium*, and which may be of independent interest. We show that affine equilibria cannot be exploited by any Nash strategy of the opponent, so an opponent who wishes to exploit must open herself to counter-exploitation. Even without the safety-guaranteeing additions, experiments on medium-sized games show that our approach always reduced exploitability in practical games even when applied at every infoset, and a depth-limited version of it led to—to our knowledge—the first strong AI for the challenge problem *dark chess*.

## 1 Introduction

*Subgame solving* is the standard technique for playing perfect-information games that has been used by strong agents in a wide variety of games, including chess [7, 25] and go [22]. Methods for subgame solving in perfect-information games exploit the fact that a solution to a subgame can be computed independently of the rest of the game. However, this condition fails in the imperfect-information setting, where the optimal strategy in a subgame can depend on strategies outside that subgame.

Recently, subgame solving techniques have been extended to imperfect-information games [9, 13]. Some of those techniques are provably *safe* in the sense that, under reasonable conditions, incorporating them into an agent cannot make the agent more exploitable [6, 21, 2, 20, 5, 26, 1, 16].

35th Conference on Neural Information Processing Systems (NeurIPS 2021).

These techniques formed the core ingredient toward recent superhuman breakthroughs in AIs for no-limit Texas hold'em poker [3, 4]. However, all of the prior techniques have a shared weakness that limits their applicability: as a first step, they enumerate the entire *common-knowledge closure* of the player's current infoset, which is the smallest set of states within which it is common knowledge that the current node lies. In two-player community-card poker (in which each player is dealt private hole cards, and all actions are public, e.g., Texas hold'em), for example, the common-knowledge closure contains one node for each assignment of hole cards to both players. This set has a manageable size in such poker games, but in other games, it is unmanageably large.

We introduce a different technique to avoid having to enumerate the entire common-knowledge closure. We enumerate only the set of nodes corresponding to $k$th-order knowledge for finite $k$—in the present work, we focus mostly on the case $k = 1$, for it already gives us interesting results. This allows an agent to only conduct subgame solving on still-reachable states, which in general is a much smaller set than the whole common-knowledge subgame.

We prove that, as is, the resulting algorithm, 1-KLSS, does not guarantee safety, but we develop three avenues by which safety can be guaranteed. First, safety is guaranteed if the results of subgame solves are incorporated back into the blueprint strategy. Second, we provide a method by which safety is achieved by limiting the infosets at which subgame solving is performed. Third, we prove that our approach, when applied at every infoset reached during play, achieves a weaker notion of equilibrium, which we coin *affine equilibrium* and which may be of independent interest. We show that affine equilibria cannot be exploited by any Nash strategy of the opponent: an opponent who wishes to exploit an affine equilibrium must open herself to counter-exploitation. Even without these three safety-guaranteeing additions, experiments on medium-sized games show that 1-KLSS always reduced exploitability in practical games even when applied at every infoset.

We use depth-limited 1-KLSS to create, to our knowledge, the first agent capable of playing *dark chess*, a large imperfect-information variant of chess with similar game tree size, at a high level. We test it against opponents of various levels, including a baseline agent, an amateur-level human, and the world's highest-rated player. Our agent defeated the former two handily, and, despite losing to the top human, exhibited strong performance in the opening and midgame, often gaining a significant advantage before losing it in the endgame.

## 2   Notation and definitions

An *extensive-form perfect-recall zero-sum game with explicit observations* (hereafter *game*) $\Gamma$ between two players $\oplus$ and $\ominus$ consists of:

(1) a tree $H$ of *nodes* with labeled edges, rooted at a *root node* $\emptyset \in H$. The set of leaves, or *terminal nodes*, of $H$ will be denoted $Z$. The labels on the edges are called *actions*. The child node reached by playing action $a$ at node $h$ will be denoted $ha$.

(2) a *utility function* $u : Z \to \mathbb{R}$.

(3) a map $P : (H \setminus Z) \to \{\text{NATURE}, \oplus, \ominus\}$ denoting which player's turn it is.

(4) for each player $i \in \{\oplus, \ominus\}$, and each internal node $h \in H \setminus Z$, an *observation* $\mathcal{O}_i(h)$ that player $i$ learns upon reaching $h$. The observation must uniquely determine whether player $i$ has the move; i.e., if $\mathcal{O}_i(h) = \mathcal{O}_i(h')$, then either $P(h), P(h') = i$, or $P(h), P(h') \neq i$.

(5) for each node $h$ with $P(h) = \text{NATURE}$, a distribution $p(\cdot|h)$ over the actions at $h$.

A player $i$'s *observation sequence* (hereafter *sequence*) mid-playthrough is the sequence of observations made and actions played by $i$ so far. The set of sequences of player $i$ will be denoted $\Sigma_i$. The observation sequence at node $h$ (immediately after $i$ observes $\mathcal{O}_i(h)$) will be denoted $s_i(h)$.

We say that two states $h = \emptyset a_1 \ldots a_t$ and $h' = \emptyset b_1 \ldots b_t$ are *indistinguishable to player $i$*, denoted $h \sim_i h'$, if $s_i(h) = s_i(h')$. An equivalence class of nodes $h \in H$ under $\sim_i$ is an *information set*, or *infoset* for player $i$. If two nodes at which player $i$ moves belong to the same infoset $I$, the same set of actions must be available at $h$ and $h'$. If $a$ is a legal action at $I$, we will use $Ia$ to denote the sequence reached by playing action $a$ at $I$.

If $u, u'$ are nodes or sequences, $u \preceq u'$ means $u$ is an ancestor or prefix (respectively) of $u'$ (or $u' = u$). If $S$ is a set of nodes, $h \succeq S$ means $h \succeq h'$ for some $h' \in S$, and $\overline{S} = \{z : z \succeq S\}$.

A *sequence-form mixed strategy* (hereafter *strategy*) of player $i$ is a vector $x \in \mathbb{R}^{\Sigma_i}$, in which $x(s)$ denotes the probability that player $i$ plays all the actions in the sequence $s$. If $h$ is a node or infoset, then we will use the overloaded notation $x(h) := x(s_i(h))$. The set of valid strategies for each player forms a convex polytope [15], which we will denote $X$ and $Y$ for $\oplus$ and $\ominus$ respectively. A strategy profile $(x, y) \in X \times Y$ is a pair of strategies. The payoff for $\oplus$ in a strategy profile $(x, y)$ will be denoted $u(x, y) := \sum_{z \in Z} u(z)p(z)x(z)y(z)$, where $p(z)$ is the probability that nature plays all the strategies on the path from $\emptyset$ to $z$. (The payoff for $\ominus$ is $-u(x, y)$ since the game is zero-sum.) The *payoff matrix* is the matrix $A \in \mathbb{R}^{\Sigma_\oplus \times \Sigma_\ominus}$ whose bilinear form is the utility function, that is, for which $\langle x, Ay \rangle = u(x, y)$. Most common game-solving algorithms, such as linear programming [15], counterfactual regret minimization and its modern variants [31, 8], and first-order methods such as EGT [12, 17] work directly with the payoff matrix representation of the game.

The *counterfactual best-response value* (hereafter *best-response value*) $u^*(x|Ia)$ to a $\oplus$-strategy $x \in X$ upon playing action $a$ at $I$ is the normalized best value for $\ominus$ against $x$ after playing $a$ at $I$: $u^*(x|Ia) = \frac{1}{\sum_{h \in I} p(h)x(h)} \min_{y \in Y : y(Ia) = 1} \sum_{z : s_\ominus(z) \succeq Ia} u(z)p(z)x(z)y(z)$. The best-response value at an infoset $I$ is defined as $u^*(x|I) = \max_a u^*(x|Ia)$. The *best-response value* $u^*(x)$ (without specifying an infoset) is the best-response value at the root, i.e., $\min_{y \in Y} u(x, y)$. Analogous definitions hold for $\ominus$-strategy $y$ and $\oplus$-infoset $I$. A player is playing an $\varepsilon$-*best response* in a strategy profile $(x, y)$ if $u(x, y)$ is within $\varepsilon$ of the best-response value of her opponent's strategy. We say that $(x, y)$ is an $\varepsilon$-*Nash equilibrium* ($\varepsilon$-NE) if both players are playing $\varepsilon$-best responses. *Best responses* and *Nash equilibria* are, respectively, 0-best responses and 0-Nash equilibria. An *NE strategy* is one that is part of an NE. The set of NE strategies is also a convex polytope [15].

We say that two nodes $h$ and $h'$ are *transpositions* if an observer who begins observing the game at $h$ or $h'$ and sees both players' actions and observations at every timestep cannot distinguish between the two nodes. Formally, $h, h'$ are transpositions if, for all action sequences $a_1 \ldots a_t$:

(1) $ha_1 \ldots a_t$ is valid (i.e., for all $j$, $a_j$ is a legal move in $ha_1 \ldots a_{j-1}$) if and only if $h'a_1 \ldots a_t$ is valid, and in this case, we have $\mathcal{O}_i(ha_1 \ldots a_j) = \mathcal{O}_i(h'a_1 \ldots a_j)$ for all players $i$ and times $0 \leq j \leq t$, and

(2) $ha_1 \ldots a_t$ is terminal if and only if $h'a_1 \ldots a_t$ is terminal, and in this case, we have $u(ha_1 \ldots a_t) = u(h'a_1 \ldots a_t)$.

For example, ignoring draw rules, two chess positions are transpositions if they have equal piece locations, castling rights, and *en passant* rights.

## 3 Common-knowledge subgame solving

In this section we discuss prior work on subgame solving. First, $\oplus$ computes a blueprint strategy $x$ for the full game. During a playthrough, $\oplus$ reaches an infoset $I$, and would like to perform subgame solving to refine her strategy for the remainder of the game. All prior subgame solving methods that we are aware of require, as a first step, constructing [6, 21, 2, 20, 5, 26, 1, 16], or at least approximating via samples [27], the *common-knowledge closure* of $I$.

**Definition 1.** The *infoset hypergraph* $G$ of a game $\Gamma$ is the hypergraph whose vertices are the nodes of $\Gamma$, and whose hyperedges are information sets.

**Definition 2.** Let $S$ be a set of nodes in $\Gamma$. The *order-$k$ knowledge set* $S^k$ is the set of nodes that are at most distance $k - 1$ away from $S$ in $G$. The *common-knowledge closure* $S^\infty$ is the connected component of $G$ containing $S$.

Intuitively, if we know that the true node is in $S$, then we know that the opponent knows that the true node is in $S^2$, we know that the opponent knows that we know that the true node is in $S^3$, etc., and it is common knowledge that the true node is in $S^\infty$. After constructing $I^\infty$ (where $I$, as above, is the infoset $\oplus$ has reached), standard techniques then construct the subgame $\overline{I^\infty}$ (or an abstraction of it), and solve it to obtain the refined strategy. In this section we describe three variants: *resolving* [6], *maxmargin* [21], and *reach subgame solving* [2].

Let $H_{\text{top}}$ be the set of root nodes of $I^\infty$, that is, the set of nodes $h \in I^\infty$ for which the parent of $h$ is not in $I^\infty$. In *subgame resolving*, the following gadget game is constructed. First, nature chooses a node $h \in H_{\text{top}}$ with probability proportional to $p(h)x(h)$. Then, $\ominus$ observes her infoset $I_\ominus(h)$,

and is given the choice to either *exit* or *play*. If she exits, the game ends at a terminal node $z$ with $u(z) = u^*(x|I_\ominus(h))$. This payoff is called the *alternate payoff* at $I_\ominus(h)$. Otherwise, the game continues from node $h$. In *maxmargin* solving, the objective is changed to instead find a strategy $x'$ that maximizes the minimum *margin* $M(I) := u^*(x'|I) - u^*(x|I)$ associated with any $\ominus$-infoset $I$ intersecting $H_{\text{top}}$. (Resolving only ensures that all margins are positive). This can be accomplished by modifying the gadget game. In *reach subgame solving*, the alternative payoffs $u^*(x|I)$ are decreased by the *gift* at $I$, which is a lower bound on the magnitude of error that $\ominus$ has made by playing to reach $I$ in the first place. Reach subgame solving can be applied on top of either resolving or maxmargin.

The full game $\Gamma$ is then replaced by the gadget game, and the gadget game is resolved to produce a strategy $x'$ that $\oplus$ will use to play to play after $I$. To use nested subgame solving, the process repeats when another new infoset is reached.

## 4    Knowledge-limited subgame solving

In this section we introduce the main contribution of our paper, *knowledge-limited subgame solving*. The core idea is to reduce the computational requirements of safe subgame solving methods by discarding nodes that are "far away" (in the infoset hypergraph $G$) from the current infoset.

Fix an odd positive integer $k$. In *order-$k$ knowledge-limited subgame solving* ($k$-*KLSS*), we *fix* $\oplus$'s strategy outside $\overline{I^k}$, and then perform subgame solving as usual. Pseudocode for all algorithms can be found in the appendix. This carries many advantages:

(1) Since $\oplus$'s strategy is fixed outside $\overline{I^k}$, $\ominus$'s best response outside $\overline{I^{k+1}}$ is also fixed. Thus, all nodes outside $\overline{I^{k+1}}$ can be pruned and discarded.

(2) At nodes $h \in \overline{I^{k+1}} \setminus \overline{I^k}$, $\oplus$'s strategy is again fixed. Thus, the payoff at these nodes is only a function of $\ominus$'s strategy in the subgame and the blueprint strategy. These payoffs can be computed from the blueprint and added to the row of the payoff matrix corresponding to $\oplus$'s empty sequence. These nodes can then also be discarded, leaving only $\overline{I^k}$.

(3) Transpositions can be accounted for if $k = 1$ and we allow a slight amount of incorrectness. Suppose that $h, h' \in I$ are transpositions. Then $\oplus$ cannot distinguish $h$ from $h'$ ever again. Further, $\ominus$'s information structure after $h$ in $\overline{I^k}$ is identical to her information structure in $h'$ in $\overline{I^k}$. Thus, in the payoff matrix of the subgame, $h$ and $h'$ induce two disjoint sections of the payoff matrix $A_h$ and $A_{h'}$ that are identical except for the top row (thanks to Item 2 above). We can thus remove one (say, at random) without losing too much. If one section of the matrix contains entries that are all not larger than the corresponding entries of the other part, then we can remove the latter part without any loss since it is weakly dominated.

The transposition merging may cause incorrect behavior (over-optimism) in games such as poker, but we believe that its effect in a game like dark chess, where information is transient at best and the evaluation of a position depends more on the actual position than on the players' information, is minor. Other abstraction techniques can also be used to reduce the size of the subgame, if necessary. We will denote the resulting gadget game $\Gamma[I^k]$.

In games like dark chess, even individual infosets can have size $10^7$, which means even $I^2$ can have size $10^{14}$ or larger. This is wholly unmanageable in real time. Further, very long shortest paths can exist in the infoset hypergraph $G$. As such, it may be difficult to even determine whether a given node is in $I^\infty$, much less expand all its nodes, even approximately. Thus, being able to reduce to $I^k$ for finite $k$ is a large step in making subgame solving techniques practical.

The benefit of KLSS can be seen concretely in the following parameterized family of games which we coin $N$-*matching pennies*. We will use it as a running example in the rest of the paper. Nature first chooses an integer $n \in \{1, \ldots, N\}$ uniformly at random. $\oplus$ observes $\lfloor n/2 \rfloor$ and $\ominus$ observes $\lfloor (n+1)/2 \rfloor$. Then, $\oplus$ and $\ominus$ simultaneously choose heads or tails. If they both choose heads, $\oplus$ scores $n$. If they both choose tails, $\oplus$ scores $N - n$. If they choose opposite sides, $\oplus$ scores $0$. For any infoset $I$ just after nature makes her move, there is no common knowledge whatsoever, so $\overline{I^\infty}$ is the whole game except for the root nature node. However, $I^k$ consists of only $\Theta(k)$ nodes.

On the other hand, in community-card poker, $I^\infty$ itself is quite small: indeed, in heads-up Texas Hold'Em, $I^\infty$ always has size at most $\binom{52}{2} \cdot \binom{50}{2} \approx 1.6 \times 10^6$ and even fewer after public cards

have been dealt. Furthermore, game-specific tricks or matrix sparsification [14, 29] can make game solvers behave as if $I^\infty \approx 10^3$. This is manageable in real time, and is the key that has enabled recent breakthroughs in AIs for no-limit Texas hold'em [20, 3, 4]. In such settings, we do not expect our techniques to give improvement over the current state of the art.

The rest of this section addresses the *safety* of KLSS. The techniques in Section 3 are *safe* in the sense that applying them at every infoset reached during play in a nested fashion cannot increase exploitability compared to the blueprint strategy [6, 21, 2]. KLSS is not safe in that sense:

**Proposition 3.** *There exists a game and blueprint for which applying 1-KLSS at every infoset reached during play increases exploitability by a factor linear in the size of the game.*

*Proof.* Consider the following game. Nature chooses an integer $n \in \{1, \ldots, N\}$, and tells $\oplus$ but not $\ominus$. Then the two players play matching pennies, with $\ominus$ winning if the pennies match. Consider the blueprint strategy for $\oplus$ that plays heads with probability exactly $1/2 + 2/N$, regardless of $n$. This strategy is a $\Theta(1/N)$-equilibrium strategy for $\oplus$. However, if maxmargin 1-KLSS is applied independently at every infoset reached, $\oplus$ will deviate to playing tails for all $n$, because she is treating her strategy at all $m \neq n$ as fixed, and the resultant strategy is more balanced. This strategy is exploitable by $\ominus$ always playing tails. $\qquad\square$

Despite the above negative example, we now give multiple methods by which we can obtain safety guarantees when using KLSS.

## 4.1 Safety by updating the blueprint

Our first method of obtaining safety is to immediately and permanently update the blueprint strategy after every subgame solution is computed. Proofs of the results in this section can be found in the appendix.

**Theorem 4.** *Suppose that whenever $k$-KLSS is performed at infoset $I$ (e.g., it can be performed at every infoset reached during play in a nested manner), and that subgame strategy is immediately and permanently incorporated into the blueprint, thereby overriding the blueprint strategy in $\overline{I}^k$. Then the resulting sequence of blueprints has non-increasing exploitability.*

To recover a full safety guarantee from Theorem 4, the blueprint—not the subgame solution—should be used during play, and the only function of the subgame solve is to update the blueprint for later use. One way to track the blueprint updates is to store the computed solutions to all subgames that the agent has ever solved. In games where only a reasonably small number of paths get played in practice (this can depend on the strength and style of the players), this is feasible. In other games this might be prohibitively storage intensive.

It may seem unintuitive that we cannot use the subgame solution on the playthrough on which it is computed, but we can use it forever after that (by incorporating it into the blueprint), while maintaining safety. This is because, if we allow the *choice of information set $I$* in Theorem 4 to depend on the opponent's strategy, the resulting strategy is exploitable due to Proposition 3. By only using the subgame solve result at later playthroughs, the choice of $I$ no longer depends on the opponent strategy at the later playthrough, so we recover a safety guarantee.

One might further be concerned that what the opponent or nature does in some playthrough of the game affects our strategy in later playthroughs and thus the opponent can learn more about, or affect, the strategy she will face in later playthroughs. However, this is not a problem. If the blueprint is an $\varepsilon$-NE, the opponent (or nature) can affect *which* $\varepsilon$-NE we will play at later playthroughs, but because we will always play from *some* $\varepsilon$-NE, we remain unexploitable.

In the rest of this section we prove forms of safety guarantees for 1-KLSS that do not require the blueprint to be updated at all.

## 4.2 Safety by allocating deviations from the blueprint.

We now show that another way to achieve safety of 1-KLSS is to carefully allocate how much it is allowed to deviate from the blueprint. Let $G'$ be the graph whose nodes are infosets for $\oplus$, and in which two infosets $I$ and $I'$ share an edge if they contain nodes that are in the same $\ominus$-infoset. In other words, $G'$ is the infoset hypergraph $G$, but with every $\oplus$-infoset collapsed into a single node.

**Theorem 5.** *Let $x$ be an $\varepsilon$-NE blueprint strategy for $\oplus$. Let $\mathcal{I}$ be an independent set in $G'$ that is closed under ancestor (that is, if $I \succeq I'$ and $I \in \mathcal{I}$, then $I' \in \mathcal{I}$). Suppose that 1-KLSS is performed at every infoset in $\mathcal{I}$, to create a strategy $x'$. Then $x'$ is also an $\varepsilon$-NE strategy.*

To apply this method safely, we may select beforehand a distribution $\pi$ over independent sets of $G'$, which induces a map $p : V(G') \to \mathbb{R}$ where $p(I) = \Pr_{\mathcal{I} \sim \pi}[I \in \mathcal{I}]$. Then, upon reaching infoset $I$, with probability $1 - p(I)$, play the blueprint until the end of the game; otherwise, run 1-KLSS at $I$ (possibly resulting in more nested subgame solves) and play that strategy instead. It is always safe to set $p(I) \leq 1/\chi(I^\infty)$ where $\chi(I^\infty)$ denotes the chromatic number of the subgraph of $G'$ induced by the infosets in the common-knowledge closure $I^\infty$. For example, if the game is perfect information, then $G'[I^\infty]$ is the trivial graph with only one node $I$, so, as expected, it is safe to set $p(I) = 1$, that is, perform subgame solving everywhere.

## 4.3 Affine equilibrium, which guarantees safety against all equilibrium strategies.

We now introduce the notion of *affine equilibrium*. We will show that such equilibrium strategies are safe against all NE strategies, which implies that they are only exploitable by playing non-NE strategies, that is, by opening oneself up to counter-exploitation. We then show that 1-KLSS finds such equilibria.

**Definition 6.** A vector $x$ is an *affine combination* of vectors $x_1, \ldots, x_k$ if $x = \sum_{i=1}^k \alpha_i x_i$ with $\sum_i \alpha_i = 1$, where the coefficients $\alpha_i$ can have arbitrary magnitude and sign.

**Definition 7.** An *affine equilibrium strategy* is an affine combination of NE strategies.

In particular, if the NE is unique, then so is the affine equilibrium. Before stating our safety guarantees, we first state another fact about affine equilibria that illuminates their utility.

**Proposition 8.** *Every affine equilibrium is a best response to every NE strategy of the opponent.*

In other words, every affine equilibrium is an NE of the restricted game $\Gamma'$ in which $\ominus$ can only play her NE strategies in $\Gamma$. That is, affine equilibria are not exploitable by NE strategies of the opponent, not even by safe exploitation techniques [10]. So, the only way for the opponent to exploit an affine equilibrium is to open herself up to counter-exploitation. Affine equilibria may be of independent interest as a reasonable relaxation of NE in settings where finding an exact or approximate NE strategy may be too much to ask for.

**Theorem 9.** *Let $x$ be a blueprint strategy for $\oplus$, and suppose that $x$ happens to be an NE strategy. Suppose that we run 1-KLSS using the blueprint $x$, at every infoset in the game, to create a strategy $x'$. Then $x'$ is an affine equilibrium strategy.*

The theorem could perhaps be generalized to approximate equilibria, but the loss of a large factor (linear in the size of the game, in the worst case) in the approximation would be unavoidable: the counterexample in the proof of Proposition 3 has a $\Theta(1/N)$-NE becoming a $\Theta(1)$-NE, in a game where the Nash equilibria are already affine-closed (that is, all affine combinations of Nash equilibria are Nash equilibria). Furthermore, it is nontrivial to even define $\varepsilon$-affine equilibrium.

Theorem 9 and Proposition 3 together suggest that 1-KLSS may make mistakes when $x$ suffers from *systematic* errors (e.g., playing a certain action $a$ too frequently *overall* rather than in a particular infoset). 1-KLSS may overcorrect for such errors, as the counterexample clearly shows. Intuitively, if the blueprint plays action $a$ too often (e.g., folds in poker), 1-KLSS may try to correct for that game-wide error fully in each infoset, thereby causing the strategy to overall be very far from equilibrium (e.g., folding way too infrequently in poker). However, we will demonstrate that this overcorrection never happens in our experiments in practical games, even if the blueprint contains very systematic errors.

Strangely, the proofs of both Theorem 9 and Theorem 5 do not work for $k$-KLSS when $k > 1$, because it is no longer the case that the strategies computed by subgame solving are necessarily played—in particular, for $k > 1$, $k$-KLSS on an infoset $I$ computes strategies for infosets $I'$ that are no longer reachable, and such strategies may never be played. For $k = \infty$—that is, for the case of common knowledge—it is well known that the theorems hold via different proofs [6, 21, 2]. We leave the investigation of the case $1 < k < \infty$ for future research.

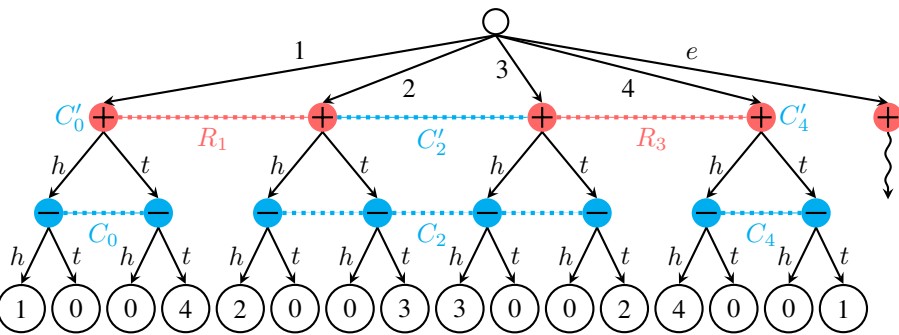

Figure 1: A simple game that we use in our example. The game is a modified version of 4-matching pennies. The two players are red ($\oplus$) and cyan ($\ominus$). Fill color of a node indicates the player to move at that node. Blank nodes are nature or terminal; terminal nodes are labeled with their utilities. Nodes will be referred to by the sequence of edges leading to that node; for example, the leftmost terminal node is $1hh$. Dotted lines indicate information sets for that player, and the colored labels give names to those information sets ($R$ for red and $C$ for cyan). (The $\ominus$-infosets $C_0'$ and $C_4'$ are singletons, containing nodes 1 and 4 respectively). The details of the subgame at $e$ are irrelevant. Nature's strategy at the root node is uniform random.

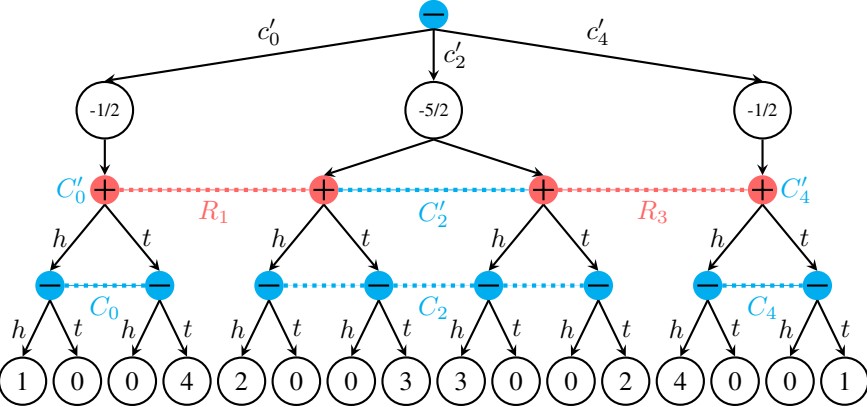

Figure 2: The common-knowledge subgame at $R_1$, $\Gamma[R_1^\infty]$. Nature's strategy at all its nodes, once again, is uniform random. The nodes $c_0'$ and $c_4'$ are redundant because nature only has one action, but we include these for consistency with the pseudocode.

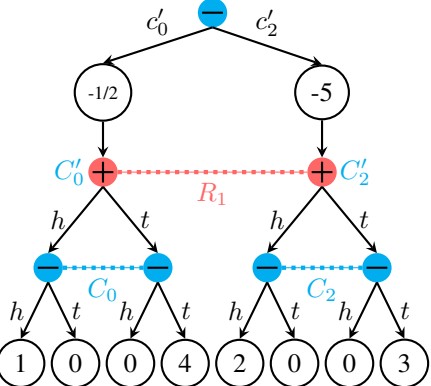

Figure 3: The subgame for 1-KLSS at $R_1$. Once again, both nature nodes are redundant, but included for consistency with the pseudocode. The counterfactual value at $c_2'$ is scaled up because the other half of the subtree is missing. In addition to this, $\ominus$ gains value $3/2$ for playing $h$ and 1 for playing $t$ at $C_2$, accounting for that missing subtree.

# 5 Example of how 1-KLSS works

Figure 1 shows a small example game. Suppose that the $\oplus$-blueprint is uniform random, and consider an agent who has reached infoset $R_1$ and wishes to perform subgame solving. Under the given blueprint strategy, $\ominus$ has the following counterfactual values: $1/2$ at $C_0'$ and $C_4'$, and $5/2$ at $C_2'$.

The common-knowledge maxmargin gadget subgame $\Gamma[R_1^\infty]$ can be seen in Figure 2. The 1-KLSS maxmargin gadget subgame $\Gamma[R_1]$ can be seen in Figure 3.

The advantage of 1-KLSS is clearly demonstrated in this example: while both KLSS and common-knowledge subgame solving prune out the subgame at node 5, 1-KLSS further prunes the subgames at node 4 (because it is outside the order-2 set $R_1^2$ and thus does not directly affect $R_1$) and node 3 (because it only depends on $\ominus$'s strategy in the subgame—and not on $\oplus$'s strategy—and thus can be added to a single row of $B$).

The payoff matrices corresponding to these gadget subgames can be found in Appendix C.

# 6 Dark chess: An agent from only a value function rather than a blueprint

In this section, we give an overview of our dark chess agent, which uses 1-KLSS as a core ingredient. More details can be found in Appendix D. Although we wrote our agent in a game-specific fashion, many techniques in this section also apply to other games.

**Definition 10.** A *trunk* of a game $\Gamma$ is a modified version of $\Gamma$ in which some internal nodes $h$ of $\Gamma$ have been replaced by terminal nodes and given utilities. We will call such nodes *internal leaves*. When working with a trunk, internal leaves $h$ can be *expanded* by adding all of their children into the tree, giving these children utilities, and removing the utility assigned to $h$.

In dark chess, constructing a blueprint is already a difficult problem due to the sheer size of the game, and expanding the whole game tree is clearly impractical. Instead, we resort to a *depth-limited* version of 1-KLSS. In depth-limited subgame solving, only a trunk of the game tree is expanded explicitly, and approximations are made to the leaves of the trunk.

Conventionally in depth-limited subgame solving of imperfect-information games, at each trunk leaf, both players are allowed to choose among *continuation strategies* for the remainder of the game [5, 1, 16, 27]. In the absence of a mechanism for creating a reasonable blueprint, much less multiple blueprints to be used as continuation strategies, we resort to only using an approximate value function $\tilde{u} : H \to \mathbb{R}$. We will not formally define what a good value function is, except that it should roughly approximate "the value" of a node $h \in H$, to the extent that such a quantity exists (for a more rigorous treatment of value functions in subgame solving, see Kovařík et al., 2021 [16]). In this setting, this is not too bothersome: the dominant term in any reasonable node-value function in dark chess will be material count, which is common knowledge anyway. We use a value function based on *Stockfish 13*, currently the strongest available chess engine.

Subgame solving in imperfect-information games with only approximate leaf values (and no continuation strategies) has not been explored to our knowledge (since it is not theoretically sound), but it seems reasonable to assume that it would work well with sufficient depth, since increasing depth effectively amounts to adding more and more continuation strategies.

To perform nested subgame solving, every time it is our turn, we perform 1-KLSS at our current information set. The generated subgame then replaces the original game, and the process repeats. This approach has the notable problem of information loss over time: since all the solves are depth-limited, eventually, we will reach a point where we fall off the end of the initially-created game tree. At this point, those nodes will disappear from consideration. From a game-theoretic perspective, this equates to always assuming that the opponent knew the exact state of the game $d$ timesteps ago, where $d$ is the search depth. As a remedy, one may consider sampling some number of infosets $I' \succeq I^2 \setminus I$ to continue expanding. We do not investigate this possibility here, as we believe that it would not yield a significant performance benefit in dark chess (and may even hurt in practice: since no blueprint is available at $I'$, a new blueprint would have to be computed. This effectively amounts to 3-KLSS, which may lack theoretical guarantees compared to 1-KLSS).

Table 1: Experimental results in medium-sized games. Reward ranges in all games were normalized to lie in $[-1, 1]$. *Ratio* is the blueprint exploitability divided by the post-subgame-solving exploitability. The value $\varepsilon$ was set to $0.25$ in all experiments, but the results are qualitatively similar with smaller values of $\varepsilon$ such as $0.1$. In the $\varepsilon$-bet/fold variants, the blueprint is the least-exploitable strategy that always plays that action with probability at least $\varepsilon$ (Kuhn poker with $0.25$-fold has an exact Nash equilibrium for P1, so we do not include it). Descriptions and statistics about the games can be found in the appendix.

| | exploitability | | |
|---|---|---|---|
| game | blueprint | after 1-KLSS | ratio |
| 2x2 Abrupt Dark Hex | .0683 | .0625 | 1.093 |
| 4-card Goofspiel, random order | .171 | .077 | 2.2 |
| 4-card Goofspiel, increasing order | .17 | .0 | $\infty$ |
| Kuhn poker | .0124 | .0015 | 8.3 |
| Kuhn poker ($\varepsilon$-bet) | .0035 | .0 | $\infty$ |
| 3-rank limit Leduc poker | .0207 | .0191 | 1.087 |
| 3-rank limit Leduc poker ($\varepsilon$-fold) | .0065 | .0057 | 1.087 |
| 3-rank limit Leduc poker ($\varepsilon$-bet) | .0097 | .0096 | 1.011 |
| Liar's Dice, 5-sided die | .181 | .125 | 1.45 |
| 100-Matching pennies | .0013 | .0098 | 0.13 |

## 7   Experiments

**Experiments in medium-sized games.** We conducted experiments on various small and medium-sized games to test the practical performance of 1-KLSS. To do this, we created a blueprint strategy for $\oplus$ that is intentionally weak by forcing $\oplus$ to play an $\varepsilon$-uniform strategy (i.e., at every infoset $I$, every action $a$ must be played with probability at least $\varepsilon/m$ where $m$ is the number of actions at $I$). The blueprint is computed as the least-exploitable strategy under this condition. During subgame solving, the same restriction is applied at every infoset except the root, which means theoretically that it is possible for any strategy to arise from nested solving applied to every infoset in the game. The mistakes made by playing with this restriction are highly systematic (namely, playing bad actions with positive probability $\varepsilon$); thus, the argument at the end of Section 4 suggests that we may expect order-1 subgame solving to perform poorly in this setting.

We tested on a wide variety of games, including some implemented in the open-source library *OpenSpiel* [19]. All games were solved with Gurobi 9.0 [11], and subgames were solved in a nested fashion at every information set using *maxmargin* solving. We found that, in all practical games (i.e., all games tested except the toy game 100-matching pennies) 1-KLSS in practice always decreases the exploitability of the blueprint, suggesting that 1-KLSS decreases exploitability in practice, despite the lack of matching theoretical guarantees. Experimental results can be found in Table 1. We also conducted experiments at $\varepsilon = 0$ (so that the blueprint is an exact NE strategy, and all the subgame solving needs to do is not inadvertently ruin the equilibrium), and found that, in all games tested, the equilibrium strategy was indeed not ruined (that is, exploitability remained 0). Gurobi was reset before each subgame solution was computed, to avoid warm-starting the subgame solution at equilibrium.

The experimental results suggest that despite the behavior of 1-KLSS in our counterexample to Proposition 3, in practice 1-KLSS can be applied at every infoset without increasing exploitability despite lacking theoretical guarantees.

**Experiments in dark chess.** We used the techniques of Appendix D to create an agent capable of playing dark chess. We tested on dark chess instead of other imperfect-information chess variants, such as *Kriegspiel* or *recon chess*, because dark chess has recently been implemented by a major chess website, chess.com (under the name *Fog of War Chess*), and has thus exploded in recent popularity, producing strong human expert players. Our agent runs on a single machine with 6 CPU cores.

We tested our agent by playing three different opponents:

(1) A 100-game match against a baseline agent, which is, in short, the same algorithm as our agent, except that it only performs imperfect-information search to depth 1, and after that uses *Stockfish*'s perfect-information evaluation with iterative deepening. The baseline agent

is described in more detail the appendix. Our agent defeated it by a score of 59.5–40.5, which is statistically significant at the 95% level.

(2) One of the authors of this paper is rated approximately 1700 on chess.com in Fog of War, and has played upwards of 20 games against the agent, winning only two and losing the remainder.

(3) Ten games against FIDE Master Luis Chan ("luizzy"), who is currently the world's strongest player on the Fog of War blitz rating list[1] on chess.com, with a rating of 2416. Our agent lost the match 9–1. Despite the loss, our agent demonstrated strong play in the opening and midgame phases of the game, often gaining a large advantage before throwing it away in the endgame by playing too pessimistically.

The performances against the two humans put the rating of our agent at approximately 2000, which is a strong level of play. The agent also exhibited nontrivial plays such as bluffing by attacking with unprotected pieces, and making moves that exploit the opponent's lack of knowledge—something that agents like the baseline agent could never do. We have compiled and uploaded some representative samples of gameplay of our dark chess agent, with comments, at this link.

## 8    Conclusions and future research

We developed a novel approach to subgame solving, $k$-KLSS, in imperfect-information games that avoids dealing with the common-knowledge closure. Our methods vastly increase the applicability of subgame solving techniques; they can now be used in settings where the common-knowledge closure is too large to enumerate or approximate. We proved that as is, this does not guarantee safety of the strategy, but we developed three avenues by which safety guarantees can be achieved. First, safety is guaranteed if the results of subgame solves are incorporated back into the blueprint strategy. Second, the usual guarantee of safety against *any* strategy can be achieved by limiting the infosets at which subgame solving is performed. Third, we proved that 1-KLSS, when applied at every infoset reached during play, achieves a weaker notion of equilibrium, which we coin *affine equilibrium* and which may be of independent interest. We showed that affine equilibria cannot be exploited by any Nash strategy of the opponent, so an opponent who wishes to exploit an affine equilibrium must open herself to counter-exploitation. Even without the safety-guaranteeing additions, experiments on medium-sized games showed that 1-KLSS always reduced exploitability in practical games even when applied at every infoset, and depth-limited 1-KLSS led to, to our knowledge, the first strong AI for dark chess.

This opens many future research directions:

(1) Analyze $k$-KLSS for $1 < k < \infty$ in theory and practice.

(2) Incorporate function approximation via neural networks to generate blueprints, particles, or both.

(3) Improve techniques for large games such as dark chess, especially managing possibly-game-critical uncertainty about the opponent's position and achieving deeper, more accurate search.

## Acknowledgments and Disclosure of Funding

This material is based on work supported by the National Science Foundation under grants IIS-1718457, IIS-1901403, and CCF-1733556, and the ARO under award W911NF2010081. We also thank Noam Brown and Sam Sokota for helpful comments.

---

[1]That rating list is by far the most active, so it is reasonable to assume those ratings are most representative.

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
