# A Proofs

We start with a lemma that we will repeatedly use in the proofs.

**Lemma 11.** *Let $(x, y)$ be a blueprint strategy, and $I$ be an infoset for player 1 with $x(I) > 0$. Then fixing strategies for both players at all nodes $h \not\sqsupseteq I$; performing resolving, maxmargin, or reach subgame solving at only $\overline{I^k}$; and then playing according to that strategy in $\overline{I^k}$ and $x$ elsewhere, results in a strategy $x'$ that is not more exploitable than $x$.*

*Proof.* Identical to the proof of safety of subgame resolving [6]: we always have access to our blueprint strategy, which by design makes all margins nonnegative. □

## A.1 Proposition 8

Let $y^*$ be a $\ominus$-NE strategy. Let $x$ be an affine equilibrium for $\oplus$, and write $x = \sum_i \alpha_i x_i^*$ where $x_i^*$ are Nash equilibria, and $\sum_i \alpha_i = 1$ (but $\alpha_i$ are not necessarily positive). Then we have

$$u(x, y^*) = \sum_i \alpha_i u(x_i^*, y^*) = u^*. \qquad \square$$

## A.2 Theorem 4

Apply Lemma 11 repeatedly. □

## A.3 Theorem 5

By induction on the infoset structure. Assume WLOG that $\oplus$ has a root infoset $I_0$.

*Base case.* If $\oplus$ has only one infoset, then Lemma 11 applies.

*Inductive case.* Let $\mathcal{I}' \subset \mathcal{I}_1$ be the collection of infosets that could be the next infosets reached after $I_0$. Formally, $\mathcal{I}' = \{I \in \mathcal{I}_1 : I \succ I_0 \text{ and there is no } I' \text{ such that } I \succ I' \succ I_0\}$. Since $\mathcal{I}$ is closed under ancestors, for each infoset $I \in \mathcal{I}' \setminus \mathcal{I}$, the downward closure $\bar{I}$ does not intersect with $\mathcal{I}$. Thus, the strategy in $\bar{I}$ will be left untouched, and is treated as fixed by all subgame solves.

Subgame solving is then performed at every information set $I \in \mathcal{I} \cap \mathcal{I}'$. By inductive hypothesis, for each $I$, this gives a Nash equilibrium $x_I$ of $\Gamma[I]$, which, by definition of $\Gamma[I]$, makes all margins in that subgame nonnegative. Since $\mathcal{I}$ is an independent set, the margin of each $\ominus$-infoset is only dependent on at most one of the subgame solves. Thus, replacing the strategy in $\bar{I}$ with $x_I$ for each $I \in \mathcal{I} \cap \mathcal{I}'$ still leaves all nonnegative margins in the original game, which completes the proof. □

## A.4 Theorem 9

By induction on the infoset structure. As above, assume WLOG that $\oplus$ has a root infoset $I_0$.

*Base case.* If $\oplus$ has only one infoset, then Lemma 11 applies.

*Inductive case.* Let $\mathcal{I}'$ be as in the previous proof. By inductive hypothesis, for each $I \in \mathcal{I}'$, running subgame solving on $\bar{I}$ yields a strategy $x_I$ that is an affine equilibrium in $\Gamma[I]$. By definition of affine equilibrium, write $x_I = \sum_j \alpha_{I,j} x_{I,j}$ where $x_{I,j}$ are Nash equilibria of $\Gamma[I]$. Let $x'_I$ be the strategy in $\Gamma$ defined by playing according to $x_I$ in $\bar{I}$, and the blueprint everywhere else.

Then each $x'_I$ is an affine equilibrium, because it is an affine combination of the strategies $x'_{I,j}$, which by Lemma 11 are Nash equilibria of $\Gamma$. But then the strategy created by running subgame solving at every $I \in \mathcal{I}'$, which is $x + \sum_{I \in \mathcal{I}'}(x'_I - x)$, is an affine combination of affine equilibria, and hence itself an affine equilibrium. □

# B Description of games

## B.1 Dark chess

Imperfect information games model real-world situations much more accurately than perfect-information games. Imperfect-information variants of chess include *Kriegspiel*, *recon chess*, and

*dark chess*. Nowadays, by far the most popular of the variants is *dark chess*, because it has been implemented by the popular chess website chess.com, and strong human experts have emerged. We thus focus on this variant as a benchmark.

*Dark chess*, also known as *fog of war chess* on chess.com, is like chess, except with the following modifications:

(1) Each player only observes the squares that her own pieces can legally move to.

(2) A player knows what squares she can see. In particular, if a pawn is blocked from moving forward by an opponent piece, the player knows that the pawn is blocked but does not know what piece is the blocker (unless, of course, another piece can see the relevant square).

(3) If there is a legal en-passant capture, the player is told the en-passant square.

(4) There is no check or checkmate. The objective of the game is to capture the opposing king. Thus, in particular, "stalemate" is a forced win for the stalemating player, and castling into, out of, or through "check" is legal (though the former, of course, loses immediately).

These rules imply that a player always knows her exact set of legal moves. As in standard chess, the game is drawn on three-fold repetition, or 50 full moves without any pawn move or capture (Unlike in standard chess, it is up to the game implementation to declare a draw, since the players may not know about the 50-move counter or past repetitions).

For purposes of determining transpositions, our agent ignores draw rules. If a node $h$ *could be* drawn (i.e., if we have repeated an observation three times, or have gone 50 moves without *observing* a pawn move or capture), then the value $\tilde{u}(h)$ of that node and all its descendants is capped at $0$. This way, the agent actively avoids possible draws only when winning.

## B.2 Other games used in experiments

Table 2: Game statistics of games in this subsection. The averages are taken over *nodes*; that is, they are the average size of $I^k$ for uniformly-sampled nodes $h$ in the game tree, where $I$ is the infoset containing $h$. "diam" is the diameter of the infoset hypergraph—equivalently, the smallest $k$ such that $I^k = I^\infty$ for all $I$. We note that the main purpose of the experiments on these games was to demonstrate practical safety, not necessarily to exhibit games of large diameter or in which the average common-knowledge size is necessarily large.

| game | nodes | infosets | diameter | average $\left|I^k\right|$ for $k = \ldots$ | | | | |
|---|---|---|---|---|---|---|---|---|
| | | | | 1 | 2 | 3 | 4 | $\infty$ |
| 2x2 Abrupt Dark Hex | 471 | 94 | 13 | 5.23 | 12.00 | 18.17 | 22.04 | 29.58 |
| 4-card Goofspiel, random | 26773 | 3608 | 4 | 5.84 | 8.90 | 9.19 | 9.20 | |
| 4-card Goofspiel, increasing | 1077 | 162 | 4 | 5.83 | 9.05 | 9.31 | 9.32 | |
| Kuhn poker | 58 | 12 | 3 | 2.50 | 3.50 | 4.00 | | |
| 3-rank limit Leduc poker | 9457 | 936 | 3 | 6.14 | 14.71 | 15.40 | | |
| Liar's Dice, 5-sided die | 51181 | 5120 | 2 | 7.00 | 15.00 | | | |
| 100-Matching pennies | 701 | 101 | 99 | 3.63 | 4.29 | 4.93 | 5.57 | 35.97 |

All games in this subsection, except $k$-matching pennies (which is described in the paper body), are implemented in *OpenSpiel* [19].

*Kuhn poker* [18] and *Leduc poker* [24] are small variants of poker. In Kuhn poker, each player is dealt one of three cards, and a single round of betting ensues with a fixed bet size and a one-bet limit. There are no community cards. In Leduc poker, there is a deck of six cards. Each player is dealt a hole card, and there is a single community card. There are two rounds of betting, one before and one after the community card is dealt. There is a two-bet limit per round, and the raise sizes are fixed.

*Abrupt dark hex* is the board game *Hex*, except that a player does not observe the opponent's moves. If a player attempts to play an illegal move, she is notified, and she loses her turn.

*k-card Goofspiel* is played as follows. At time $t$ (for $t = 1, \ldots, k$), players simultaneously place bids for a prize of value $v_t$. The possible bids are the integers $1, \ldots, k$. Each player must use each bid exactly once. The higher bid wins the prize; in the event of a tie, the prize is split. The players

learn who won the prize, but do not learn the exact bid played by the opponent. In the *random card order* variant, the list $\{v_t\}$ is a random permutation of $\{1, \dots, k\}$. In the *fixed increasing card order* variant, $v_t = t$.

*Liar's dice.* Two players roll independent dice. The players then alternate making claims about the value of their own die (e.g., "my die is at least 3"). Each claim must be larger than the previous one, until someone calls *liar*. If the last claim was correct, the claimant wins.

## C    Example of $1$-KLSS

We first introduce some notation that we will use in this section.

We will explicitly specify what game is in discussion using notation like $\Sigma_i^\Gamma$ to reference the set of player $i$'s sequences in game $\Gamma$. In particular, if $x^\Gamma \in \mathbb{R}^{\Sigma_i^\Gamma}$ is a strategy for player $i$, and $\Gamma'$ is a subgame of $\Gamma$, we will let $x^{\Gamma'}(s) = x(s)/x(I)$ where $I \preceq s$ is a root infoset in $\Gamma'$.

In addition to the typical payoff matrix $A^\Gamma \in \mathbb{R}^{\Sigma_\oplus^\Gamma \times \Sigma_\ominus^\Gamma}$, we will also treat games as having an explicit additional payoff matrix $B^\Gamma \in \mathbb{R}^{\Sigma_\oplus^\Gamma \times \Sigma_\ominus^\Gamma}$, so that the payoff of a strategy profile $(x, y)$ is $\langle x, (A^\Gamma + B^\Gamma)y \rangle$. The top row of $B^\Gamma$ will be used to store alternate payoffs in subgames, as well as the utility that $\ominus$ gains from nodes outside $\overline{I^k}$ (see Section 4). The first column of $B^\Gamma$ will be used to store the entropy penalties in our dark chess agent (see Appendix D). $B^\Gamma$ will be empty except for these entries.

### C.1    Common-knowledge subgame

The reward matrix $A^{\Gamma[R_1^\infty]}$ has the following entries, corresponding to terminal nodes in $\Gamma[R_1^\infty]$:

| $\oplus$ \ $\ominus$ | $\emptyset$ | $c_0'$ | $C_0h$ | $C_0t$ | $c_2'$ | $C_2h$ | $C_2t$ | $c_4'$ | $C_4h$ | $C_4t$ |
|---|---|---|---|---|---|---|---|---|---|---|
| $\emptyset$ | | | | | | | | | | |
| $R_1h$ | | | 1 | 0 | | 1 | 0 | | | |
| $R_1t$ | | | 0 | 4 | | 0 | 3/2 | | | |
| $R_3h$ | | | | | | 3/2 | 0 | | 4 | 0 |
| $R_3t$ | | | | | | 0 | 1 | | 0 | 1 |

In addition, we must subtract off $\ominus$'s counterfactual values: $1/2$ from playing $c_0'$, $5/2$ from playing $c_2'$, and $2$ from playing $c_4'$. Thus, $B^{\Gamma[R_1^\infty]}$ has the following nonzero entries:

| $\oplus$ \ $\ominus$ | $\emptyset$ | $c_0'$ | $C_0h$ | $C_0t$ | $c_2'$ | $C_2h$ | $C_2t$ | $c_4'$ | $C_4h$ | $C_4t$ |
|---|---|---|---|---|---|---|---|---|---|---|
| $\emptyset$ | | $-1/2$ | | | $-5/2$ | | | $-1/2$ | | |
| $\vdots$ | | | | | | | | | | |

### C.2    $1$-KLSS subgame

The reward matrix $A^{\Gamma[R_1]}$ has the following entries, corresponding to terminal nodes in $\Gamma[R_1]$:

| $\oplus$ \ $\ominus$ | $\emptyset$ | $c_0'$ | $C_0h$ | $C_0t$ | $c_2'$ | $C_2h$ | $C_2t$ |
|---|---|---|---|---|---|---|---|
| $\emptyset$ | | | | | | | |
| $R_1h$ | | | 1 | 0 | | 2 | 0 |
| $R_1t$ | | | 0 | 4 | | 0 | 3 |

In addition, we must subtract off $\ominus$'s counterfactual values: $1/2$ from playing $c_0'$, and $3$ from playing $c_2'$ (the reward at $c_2$ is scaled up, because the subtree at the node $3$ is missing!). Further, from the subtree at node $3$, $\ominus$ has alternate values $3/2$ at $C_2h$ and $1$ at $C_2t$. Thus, $B^{\Gamma[R_1]}$ has the following nonzero values:

| $\oplus$ ╲ $\ominus$ | $\emptyset$ | $c_0'$ | $C_0h$ | $C_0t$ | $c_2'$ | $C_2h$ | $C_2t$ |
|---|---|---|---|---|---|---|---|
| $\emptyset$ | | $-1/2$ | | | $-5$ | $3/2$ | $1$ |
| $\vdots$ | | | | | | | |

## D  Dark chess agent details

In this section, we give further details of our dark chess agent.

### D.1  Value function

For a value function, we run *Stockfish 13* on the position at depth $1$, and then clamp the reward to a range $[-\tau, \tau]$ (where $\tau$ is a tuneable hyperparameter; we set $\tau = 6$ pawns) via the mapping $x \mapsto \tanh(x/\tau)$. Using Stockfish's evaluation function saves us the trouble and resources required to learn chess from scratch, and clamping it to a finite range ensures that our agent understands that, after a certain point, a higher evaluation does not indicate a substantially higher probability of victory.

### D.2  Adapting techniques from perfect-information game solving

*Iterative deepening* is a natural approach to incrementally generate the game tree when solving a game in the perfect-information setting [23], and is used by most strong chess engines. We suggest a natural extension of iterative deepening to imperfect-information games. At all times, maintain a trunk that initially contains only the root node. Solve the trunk game exactly (e.g., with an LP solver). If time permits, expand all internal leaves that are in the support of *either* player's strategy, and repeat. This technique is sound in the sense that if it does not expand any node, then an equilibrium of the full game has been found. It carries some resemblance to recent techniques for generating *certificates* [28, 30], but unlike in that paper, we do not assume nontrivial upper bounds on internal node utilities, so we cannot expand only the nodes reached by both players.

If a reasonable *move ordering* exists over moves that approximates how "interesting" or "strong" a move is in a given position, it can be used to focus the search. Instead of expanding *all* leaves in the support of either player's strategy, we use the move ordering to judiciously pick which nodes to expand. If an internal node $h$ in the support of at least one player's strategy has multiple unexpanded children $ha$, we start by only expanding those children that are in the support of *both* players' current strategies. Of the children that are not, we expand only the child that is the most "interesting", delaying the expansion of the other children to a later iteration. For our dark chess agent, the "interestingness" of a child is defined by its estimated value $\tilde{u}(ha)$, except that checks, captures, and promotions are always defined to be more interesting than other moves. This change allows us to focus our attention on parts of the game tree that are easy for the value function $\tilde{u}$ to misunderstand—namely, positions in which there are forcing moves—thereby allowing a much deeper search.

### D.3  Dealing with lost particles

Upon reaching a new infoset $I$ in a playthrough, because we are performing non-uniform iterative deepening, it is likely that some nodes in $I$ do not appear in the subgame search tree. It is even possible that *no* node in $I$ appears in the subgame search tree. For this reason, in addition to nested subgame solving, we maintain the exact set $I$ (up to transpositions, as per Section 4). The set $I$ rarely exceeds size $10^7$, making it reasonable to maintain and update in real time. Let $I'$ be the set of game nodes currently being considered by the player. We set a lower limit $L$ on the number of "particles" (subgame root states) being considered. If $|I'| \leq L$ and $I' \subsetneq I$, then we sample at most $L - |I'|$ nodes uniformly at random without replacement from $I \setminus I'$, and add them as roots of the subgame tree. At such nodes $h$, our agent assumes that the opponent knows the exact node. The alternate payoff at $h$ is defined to be $\min(\tilde{u}(h), \hat{u})$ where $\hat{u}$ is the estimate of our current value in the game, as deduced from the previous subgame solve. This alternate payoff setting prevents the agent from over-valuing states with $\tilde{u}(h)$ values that are unattainable due to lack of information. Since this results in a highly lopsided tree (the newly-sampled root states have not been expanded at all, whereas other states may have been searched deeply), on the $d$th iteration of the iterative deepening loop, we only allow the expansion of nodes at depth at most $d$ unless those nodes are in the support

of both players' strategies. This allows the newly-sampled roots to "catch up" to the rest of the game tree in depth.

We set $L = 200$, which we find gives a reasonable balance between achievable depth in subgame solving and representative coverage of root nodes. To prevent the set $I$ from growing too large to manage, we explicitly incentivize the agent to discover information: for each action $a$ available to the agent at the root infoset of the subgame, let $\mathcal{H}(a)$ denote the binary entropy of the next observation after playing action $a$, assuming that the true root is uniformly randomly drawn from $I'$. Then we give an explicit penalty of $2^{-\mathcal{H}(a)}|I|/M$ if the agent plays action $a$, where $M$ is a tunable hyperparameter. In our experiments, we set $M = 10^7$. The only purpose of this explicit penalty is to prevent the agent from running out of memory or time trying to compute $I$; typically $|I|$ is small enough that it is a non-factor and the agent is able to seek information without much explicit incentive.

Performing particle filtering over $I^\infty$ was suggested as an alternative in parallel work [27]. We believe that particle filtering would not work as well as our method in dark chess. If we maintained $I^\infty$ instead of $I$, the $L$ particles would have to cover the entire common-knowledge closure $I^\infty$, not just $I$, which means a coarser and thus inferior approximation of $I^\infty$. In a domain like dark chess where managing one's own uncertainty of the position is a critical part of playing good moves (since good moves in chess are highly position dependent), this will degrade performance, especially when $I^\infty$ is large compared to $I$ (which will typically be the case in dark chess).

### D.4 Choice of subgame solving variant

The choice of subgame solving variant is a nontrivial one in our setting. Due to the various approximations and heuristics used, it is often impossible to make all margins positive in a subgame. Thus, we make a hybrid decision: we first attempt *reach-maxmargin* subgame solving [2], which is a generalization of maxmargin subgame solving that incorporates the fact that we can give back the gifts the opponent has given us and still be safe (Section 3)[2]. Using reach reasoning (i.e., mistakes reasoning) gives us a larger safe strategy space to optimize over and thus larger margins. If all margins in that optimization are positive, we stop. Otherwise, we use reach-resolving instead. This makes our agent *pessimistic on offense* (if margins are positive, it assumes that the opponent is able to exactly minimize the margin), and *optimistic on defense* (in the extreme case when all margins are negative, the distribution of root nodes is assumed to be uniform random). This guarantees that all margins are made positive whenever possible, and thus, that at least modulo all the approximations, the theoretical guarantees of Theorem 9 are maintained. We find that this gives the best practical performance in experiments.

## E   Pseudocode of Algorithms

In this section, we give detailed pseudocode for all variants of our subgame solving method. The pseudocode will occasionally perform operations on entries of $B^\Gamma$ that do not yet exist; in this case, the relevant information sets and sequences are added to the sequence-form representation of $\Gamma$, even if they do not contain any nodes.

We will use $\mathcal{J}_i^\Gamma$ to denote the collection of information sets of player $i$ in game $\Gamma$, and $I_i(h)$ to denote the information set of player $i$ at $h$. For an information set $I$ of a player $i$, $s_i(I)$ denotes the sequence shared by all of $I$'s nodes. We assume, without loss of generality, that every pair of information sets $I_\oplus \in \mathcal{J}_\oplus^\Gamma$ and $I_\ominus \in \mathcal{J}_\ominus$ has intersection at most one node.

Algorithm 12 shows pseudocode for a generic knowledge-limited subgame solving implementation, including optional blocks for reach subgame solving, transposition merging, and converting between maxmargin and resolving. Algorithms 13 and 14 correspond, respectively, to Theorems 4 and 5. Algorithm 15 is the pseudocode of our dark chess agent, which is adaptable to any game with similar properties.

When the algorithms stipulate that a Nash equilibrium is to be found, any suitable exact or approximate method can be used, except in Line 23 of Algorithm 15, in which an exact method (such as linear programming) is desired because the algorithm continues reasons about the support of the equilibrium.

---

[2]Because we do not know a lower bound on the gifts the opponent has given us in dark chess, we use $\sum_{I'a' \prec I} (u^*(x|I'a') - u^*(x|I))$ as a gift estimate, where the values $u^*$ are computed from the blueprint.

**Algorithm 12** Knowledge-limited subgame solving

1: **function** MAKESUBGAME(game $\Gamma$, $\oplus$-blueprint $x$ for $\Gamma$, infoset $I$, order $k$, flags OPTIONS)

2:      *▷ Makes the Maxmargin subgame. To use Resolving, use the below* MAXMARGINTORE-
         SOLVE *method to convert the output $\Gamma'$.*

3:      compute the counterfactual best response values $u^*(x|s)$ for each $\ominus$-sequence $s$

4:      compute the $k$th-order knowledge set $I^k$

5:      ALTPAY $\leftarrow$ empty dictionary mapping $\mathcal{J}_\ominus^\Gamma \to \mathbb{R}$

6:      $T \leftarrow \emptyset$                  *▷ Transposition table; only used if merging transpositions*

7:      $\Gamma' \leftarrow$ empty game

8:      create root node $\emptyset^{\Gamma'}$ in $\Gamma'$, at which $\ominus$ acts

9:      **for** each $I_0 \in \mathcal{J}_\ominus^\Gamma$ with $I_0 \cap I^k \neq \emptyset$ **do**

10:          **if** MERGETRANSPOSITIONS $\in$ OPTIONS **then**

11:              *▷ Only valid if $k = 1$. If merging transpositions, it is advisable to randomly
               shuffle the order of iteration in the main loop.*

12:              $h \leftarrow$ the lone element of $I_0 \cap I$

13:              **if** $h$ is a transposition of any $h' \in T$ **then continue**

14:              add $h$ to $T$

15:          create nature node $\emptyset^{\Gamma'} I_0$ in $\Gamma'$

16:          $D \leftarrow \sum\limits_{h \in I_0 \cap I^k} p^\Gamma(h)x(h)$              *▷ Normalization constant*

17:          **for** each $h \in I_0 \cap I^k$ **do**             *▷ Build the subtree $\overline{I_0 \cap I^k}$*

18:              copy $h$ into $\Gamma'$ as a child of $\emptyset^{\Gamma'} I_0$, with

19:              $p^{\Gamma'}(h|\emptyset^{\Gamma'} I_0) = p^\Gamma(h)x(h)/D$

20:              $\mathcal{O}_i^{\Gamma'}(h) = s_i^\Gamma(h)$ for both $i \in \{\oplus, \ominus\}$

21:          $B^{\Gamma'}[\emptyset, s_\ominus(I_0)] \leftarrow -u^*(x|I_0)$             *▷ Subtract alternate value of $I_0$*

22:          **if** REACH $\in$ OPTIONS **then** $B^{\Gamma'}[\emptyset, s_\ominus(I_0)] \leftarrow B^{\Gamma'}[\emptyset, s_\ominus(I_0)] - \hat{g}(I_0)$

23:          *▷ $\hat{g}(I_0)$ is a gift estimate. We use*

$$\hat{g}(I_0) = \sum_{I'a' : I' \in \mathcal{J}_\ominus^\Gamma, I'a' \prec I'} (u^*(x|I'a') - u^*(x|I')).$$

         *See also Brown and Sandholm [2] for alternatives and further discussion.*

24:          **for** each $I' \in \mathcal{J}_\ominus^\Gamma$ with $I' \succeq I_0$ **do**         *▷ Copy $B^\Gamma$ into $B^{\Gamma'}$, correctly scaled*

25:              $B^{\Gamma'}[\emptyset, s_\ominus(I')] \leftarrow B^{\Gamma'}[\emptyset, s_\ominus(I')] + B^\Gamma[\emptyset, s_\ominus(I')]/D$

26:          **for** each terminal node $z \in \overline{I_0} \setminus \overline{I^k}$ **do**         *▷ "Add" the nodes in $\overline{I^{k+1}} \setminus \overline{I^k}$ to $\Gamma'$*

27:              $B^{\Gamma'}[\emptyset, s_\ominus(z)] \leftarrow B^{\Gamma'}[\emptyset, s_\ominus(z)] + x(z)p^\Gamma(z)u(z)/D$

28:      **return** $\Gamma'$

29: **function** MAXMARGINTORESOLVE($\Gamma$)

30:      turn $\emptyset^\Gamma$ into a nature node at which nature plays uniformly at random

31:      **for** each child node $h$ of $\emptyset^{\Gamma'}$ **do**

32:          replace $h$ with a $\ominus$-node $h_{\text{RESOLVE}}$, at which $\ominus$ has two actions:

33:              action E (for EXIT) leads to a terminal node of value 0

34:              action P (for PLAY) leads to $h$.

35:      $B^\Gamma \leftarrow (1/N)B^\Gamma$ where $N$ is the number of children of $\emptyset^\Gamma$

36:      *▷ Ensure that $B^\Gamma$ is still normalized correctly*

37:      **return** $\Gamma$

38: **function** RESOLVETOMAXMARGIN($\Gamma$)

39:      turn $\emptyset^\Gamma$ into a $\ominus$-node

40:      **for** each child node $h$ of $\emptyset^{\Gamma'}$ **do** replace $h$ with $hP$

41:      $B^\Gamma \leftarrow NB^\Gamma$ where $N$ is the number of children of $\emptyset^\Gamma$

42:      *▷ Ensure that $B^\Gamma$ is still normalized correctly*

43:      **return** $\Gamma$

---

**Algorithm 13** Safe and nested $k$-KLSS by updating the blueprint

---

1: **maintain as state:**
2:     $\Gamma^*$ — full game
3:     $x^*$ — $\oplus$-blueprint for $\Gamma^*$ (never reset)
4:     $\Gamma$ — current subgame (reset to full game after every playthrough)
5:     $x$ — $\oplus$-strategy for $\Gamma$ (reset to blueprint after every playthrough)
6: **function** RECEIVEOBSERVATION(observation $\mathcal{O}$)
7:     $I \leftarrow \{ha : h \in I, \mathcal{O}_\oplus(ha) = \mathcal{O}\}$
8:     **if** it is not our move **then return**
9:     $\Gamma \leftarrow$ MAKESUBGAME$(\Gamma, x, I, k, \{\})$
10:     ▷ *Merging transpositions and Reach subgame solving can be used safely, but this requires some care, as described in the main paper and by Brown and Sandholm [2].*
11:     **if** using RESOLVING **then** $\Gamma \leftarrow$ MAXMARGINTORESOLVE$(\Gamma)$
12:     $(x, y) \leftarrow$ Nash equilibrium of $\Gamma$
13:     **for** each sequence $s \succeq s_\oplus(I)$ in $\Gamma^*$ **do** $x^*(s) \leftarrow x(s)x^*(I)$
14:     ▷ *Update the blueprint. This step can be skipped if we are confident that $\overline{I^2}$ will never again be reached.*

---

---

**Algorithm 14** Safe and nested $k$-KLSS by incrementally allocating deviations

---

1: **maintain as state:**
2:     $\Gamma$ — current subgame (reset to full game before each playthrough)
3:     $x$ — $\oplus$-strategy for $\Gamma$ (reset to full-game blueprint before each playthrough)
4:     RUNNINGKLSS — boolean, marking whether we can continue performing subgame solving
5:        (reset to TRUE before each playthrough)
6: **function** RECEIVEOBSERVATION(observation $\mathcal{O}$)
7:     $I \leftarrow \{ha : h \in I, \mathcal{O}_\oplus(ha) = \mathcal{O}\}$
8:     **if** it is not our move **then return**
9:     $\mathcal{I} \leftarrow$ some independent set of $G'[I^\infty]$
10:     ▷ *$G'[I^\infty]$ is the graph whose nodes are the $\oplus$-infosets in $I^\infty$, and for which there is an edge between two infosets $I$ and $I'$ if they contain nodes in the same $\ominus$-infoset. The independent set $\mathcal{I}$ can be generated by any method, including incrementally across many playthroughs if memory permits, or randomly, or both. As before, this step can be skipped if we are confident that $\overline{I^2}$ will never again be reached.*
11:     **if** $I \notin \mathcal{I}$ **then** RUNNINGKLSS = FALSE
12:     **if** RUNNINGKLSS **then**
13:         $\Gamma \leftarrow$ MAKESUBGAME$(\Gamma, x, I, k, \{\})$
14:         $(x, y) \leftarrow$ Nash equilibrium of $\Gamma$
15:         add $I$ to $\mathcal{I}$
16:     sample and play move $a \sim x(\cdot|I)$

---

---

**Algorithm 15** Nested 1-KLSS with only a value function

---

 1: **maintain as state:**
 2:    $\hat{\Gamma}$ — expanded part of current subgame (cleared before every playthrough)
 3:    $(\hat{x}, \hat{y})$ — Nash equilibrium of $\Gamma$
 4:    $I$ — full current information set (reset to $\{\emptyset\}$ before every playthrough)
 5: **hyperparameters:**
 6:    $L$ — try to maintain at least this many particles. (our implementation: 200)
 7:    $M$ — denominator on the information discovery penalty term (our implementation: $10^7$)
 8: **function** RECEIVEOBSERVATION(observation $\mathcal{O}$)
 9:    $I \leftarrow \{ha : h \in I, \mathcal{O}_{\oplus}(ha) = \mathcal{O}\}$             ▷ *Transpositions can be freely merged in $I$.*
10:    **if** it is not our move **then return**
11:    $I' \leftarrow$ find our current information set in $\Gamma$
12:    **if** $I' = \emptyset$ **then** $\hat{u} \leftarrow \infty$
13:    **else** $\hat{u} \leftarrow u^{\Gamma}(\hat{x}, \hat{y}|I')$
14:    $\Gamma \leftarrow$ MAKESUBGAME($\Gamma, x, I', 1, \{\text{MERGETRANSPOSITIONS, REACH}\}$)
15:    **if** $|I'| < L$ and $I' \neq I$ **then**
16:       $S \leftarrow$ sample of size $L - |I'|$, uniformly at random and without replacement from $I \setminus I'$
17:       **for** $h \in S$ **do**
18:          add $h$ as an internal leaf to $\Gamma$
19:          $B^{\Gamma}[\emptyset, s_{\ominus}(h)] \leftarrow -\min(\hat{u}, \tilde{u}(h))$
20:    **for** each action $a$ available at $I$ **do** $B^{\Gamma}[Ia, \emptyset] \leftarrow (1 - 2^{-\mathcal{H}(a)})|I|/M$
21:    ▷ *$\mathcal{H}(a)$ is the binary entropy of the next observation received by $\oplus$, assuming that she plays action $a$ and that the opponent distribution over $\mathcal{I}$ is uniform random.*
22:    **loop**
23:       $(x, y) \leftarrow$ Nash equilibrium of $\Gamma$
24:       **if** $u^{\Gamma}(x, y) < 0$ and $\Gamma$ is a MAXMARGIN subgame **then**
25:          ▷ *Use* MAXMARGIN *if all margins are positive; else* RESOLVE
26:          $\Gamma \leftarrow$ MAXMARGINTORESOLVE($\Gamma$)
27:          $(x, y) \leftarrow$ Nash equilibrium of $\Gamma$
28:       **else if** $u^{\Gamma}(x, y) \geq 0$ and $\Gamma$ is a RESOLVE subgame **then**
29:          $\Gamma \leftarrow$ RESOLVETOMAXMARGIN($\Gamma$)
30:          $(x, y) \leftarrow$ Nash equilibrium of $\Gamma$
31:       **if** out of time **then break**
32:       **for** each $h$ in $\Gamma$ such that at least one child $ha$ is a nonterminal leaf **do**
33:          **if** $x(h) > 0$ and $y(h) > 0$ **then**
34:             **for** each child $ha$ of $h$ **do** MAYBEEXPAND($ha$)
35:          **else**
36:             let $ha$ be the most interesting nonterminal leaf of $h$
37:             ▷ *"Most interesting" is game-specific. For dark chess, we use the child $ha$ with the highest $\tilde{u}(ha)$ value, except that we always rank captures, checks, and promotions higher than all other moves.*
38:             MAYBEEXPAND($ha$)
39:    sample and play move $a \sim x(\cdot|I')$
40: **function** MAYBEEXPAND(nonterminal leaf $h$)
41:    **if** $h$ is already expanded **then return**
42:    **if** $x(h) = y(h) = 0$ **then return** ▷ *Do not expand nodes that neither player wants to reach*
43:    $u^{\Gamma}(h) \leftarrow 0$
44:    **for** each legal action $a$ at $h$ **do** add node $ha$ to $\Gamma$ with $u^{\Gamma}(ha) = \tilde{u}(ha)$

---

# F   Checklist information

## F.1   License information

*Stockfish* is licensed under GPLv3. *OpenSpiel* is licensed under the Apache License.

## F.2   Broader impacts

The techniques we have created are very general and fundamental. AI tools like the ones in this paper can help less educated and less experienced players reach the same level as expert players, thereby making the distribution of value more fair.

A potential downside is that if the technology were only available to the privileged, that could increase unfairness. We hope to avoid this by openly publishing our work.

## F.3   Research with Human Subjects

We received IRB approval for this study. The actual approval letter has a lot of information that would violate the double blind review. We include the key sentence from that letter here:

> The *[institution name redacted for double blind review]* Institutional Review Board (IRB) has reviewed and granted APPROVAL under EXPEDITED REVIEW on 5/25/2021 per 45 CFR 46.110(7a) and 21 CFR 56.110.

We sent the following message to FIDE Master Luis Chan to enlist his participation.

> Hi Luis,
>
> We are *[author names redacted for double-blind review]*. We have developed a bot capable of playing the variant Fog of War Chess, and would like to test how it performs against the strongest human player in the world, which, according to the blitz leaderboards on chess.com, is you. Would you be willing to play a match against our bot over chess.com? The games would of course be unrated. We propose 10 games at 5+5 time control, but are open to discussion regarding the format.
>
> Please let us know if you are interested.
>
> Thank you,
>
> *[author names redacted for double-blind review]*