# OpenReview forum: "Subgame solving without common knowledge"
_NeurIPS.cc/2021/Conference — NeurIPS 2021 Spotlight_

### Official Review · Reviewer_be5C · 2021-07-15

**Rating:** 8
**Confidence:** 4

**Summary:**

This paper deals with two related problems. The more fundamental contribution is an analysis and partial algorithmic solution to the problem of subgame solving in a subclass of games with intractable large public states. The other is creation and evaluation of an AI for playing dark chess - a chess variant with imperfect information. The key solution to the first problem is to assume limited nesting of player’s beliefs about the opponent’s beliefs. The paper is quite thorough in exploring at least some theoretical guarantees of this approach, since finding counterexamples would not be hard. The empirical evaluation shows that the algorithm practically works much better than the theoretical guarantees suggest. For the second problem, the paper introduces an algorithm which shares the key idea with the solution to the first problem, but then adds many theoretically unsound domain-specific modifications. The paper shows that this algorithm performs well against other programs as well as moderately strong human players.


**Limitations And Societal Impact:**

The authors did not addressed the limitations and potential negative societal impact. While a generic discussion of the dangers of game theoretic research could be added, I personally do not think it is necessary for this paper.

**Main Review:**

Originality: The idea of limiting the amount of nesting in beliefs is quite straightforward and has appeared before. However, the depth of its analysis in the context of EFGs and turning it into a decent bot for playing a human-scale game is definitely novel.

Clarity: The paper is very dense and requires quite a lot of background knowledge to sully comprehend. However, I assume it is still understandable on a higher level of abstraction even without that.

One thing that was not clear to me is how exactly were the games in the experimental evaluation split to subgames. Were all subgames in the games resolved for the evaluation in Table 1?

Quality: The notation in the paper is consistent and understandable.  The propositions are believable, though I did not check all the proofs in the appendix carefully. The experiments are well designed, thorough and performed on a wider variety of games.

Significance: The problem of large public states is a natural next research challenge in generalisation of theoretically sound imperfect information game playing to arbitrary domains. This paper proposes a solution that seems to work very well in practice and provides some insight into properties of this problem. I expect this paper to inspire further research in this direction in the near future.


Minor comments:

The long lists of sometimes even 9 citations are not really helpful for understanding the presented paper, nor for knowing what the reader should find in the referenced papers. The really look like an attempt to artificially increase the number of citations in a clique and should be, in my opinion, avoided.

I like the observations at the beginning of Section 4 (lines 149-163). I think it may be useful to reference them from follow-up papers. Restructuring them o a form of a proposition / observation with a unique number within the paper would make it easier.

Theorem 9 sounds very weak as it is. Why would we run subgame solving if we already have an equilibrium. Why it is difficult to generalize it to starting with an epsilon-equilibirum?


**Time Spent Reviewing:**

4

---

> ### Author Response · Authors · 2021-08-11
> **Response to Reviewer be5C**
>
> Thank you for the review!
>
> **One thing that was not clear to me is how exactly were the games in the experimental evaluation split to subgames. Were all subgames in the games resolved for the evaluation in Table 1?**
>
> All subgames were resolved in a nested fashion. That is, at each possible “top information set” (that is, non-root information set that is not a descendant of any other non-root information set), we perform 1-KLSS, then treat the 1-KLSS gadget game as a full game and the (epsilon-uniformly weakened) solution to the gadget game as a blueprint, and recurse. This simulates the standard method of applying subgame solving in games. We will clarify this in the camera-ready version.
>
> **The long lists of sometimes even 9 citations are not really helpful for understanding the presented paper, nor for knowing what the reader should find in the referenced papers. The really look like an attempt to artificially increase the number of citations in a clique and should be, in my opinion, avoided.**
>
> That list (there's only one long list of citations, copied verbatim twice) is more for completeness than anything else--there have been many techniques for subgame solving developed over the past few years, and it is difficult to leave any one in particular out. I understand the concern, though--in the camera-ready version, we will revise to better describe exactly why we cite the papers we do. Every paper in the list is relevant and cited elsewhere in our paper in a more specific context.
>
> **I like the observations at the beginning of Section 4 (lines 149-163). I think it may be useful to reference them from follow-up papers. Restructuring them o a form of a proposition / observation with a unique number within the paper would make it easier.**
>
> We will do this in the camera-ready version.
>
> **Theorem 9 sounds very weak as it is. Why would we run subgame solving if we already have an equilibrium. Why it is difficult to generalize it to starting with an epsilon-equilibirum?**
>
> We acknowledge that Thm 9 is rather weak (indeed, if you already knew you had a Nash blueprint, you wouldn't run subgame solving), but the theorem statement serves as an illustration of (roughly) what our method achieves in lieu of not having the type of safety guarantee that prior safe subgame solving techniques (which are unscalable for this problem) have.
>
> As we discuss in the paper, Prop 3 gives a counterexample in which 1-KLSS increases the exploitability of an epsilon-equilibrium by a linear factor in the size of the game. Any generalization of Thm 9 to the case of epsilon-equilibrium must therefore necessarily be rather weak in the general case.

---

### Official Review · Reviewer_MEQH · 2021-07-15

**Rating:** 7
**Confidence:** 5

**Summary:**

This paper proposes a novel technique for search in imperfect information games. One problem with existing subgame solving techniques is that they construct a gadget game with a chance node that samples over all possible states in the subgame. This gadget game will be prohibitively large in large games. To overcome this problem, the authors create a smaller gadget game where only states corresponding to the k-th order common knowledge. While an unsafe method as-is, the authors introduce three fixes to make this technique safe. They show empirical results on a number of openspiel games and show that k-KLSS always improves exploitability. Their main result is impressive state-of-the-art performance on dark chess.


**Limitations And Societal Impact:**

This work is mainly theoretical so I think the authors have adequately addressed the limitations and societal impact.


**Main Review:**

My main concern with this paper is that the technique is not safe. Although the authors provide theorems 4 and 5, I don’t think these theorems will change the method to make it safe. Theorem 4 is technically correct, but I don’t think it amounts to anything that will make the technique actually safe. Going back to the counterexample in proposition 3, if we play this game against 1-KLSS, we can always exploit it because we know it will always play tails. Theorem 4 doesn’t solve this problem because it only updates the blueprint after you solve the subgame and play the action. Theorem 5 also seems weak to me because it starts with the assumption that you already have an epsilon-Nash, which is a strong assumption and in which case you could just do nothing and keep an epsilon-Nash. While affine equilibrium could be a decent solution concept (I’m not quite sure), theorem 9 also assumes that you start with a Nash equilibrium blueprint. Since there are also no experiments that use or validate these theorems, I am unconvinced that they can be used to make the technique safe in practice. With that being said, I give credit to the authors for pointing out the limitations of the method, for example in the last two paragraphs of section 4.

Although this method seems to be unsafe, that does not disqualify it from being a good contribution, since previous unsafe subgame solving techniques have had good empirical success. However, since the method is unsafe, I would like to see some more empirical experiments to further validate the method. For example, in the medium-size game experiments, I am not convinced that adding epsilon-uniform noise is enough to systematically bias the blueprint. Instead, if you added epsilon-fold noise (where it folds with epsilon probability) for poker games, my guess would be that the new strategy would be worse than the old strategy. I like that many medium-sized games were used, but the results give the impression that 1-KLSS will always be less exploitable than the blueprint, which is not the case for different blueprints. In general, I would like to see more theoretical/empirical exploration of the types of games/blueprints where 1-KLSS performs well and where it will fail (such as the counterexample).

The results for Dark Chess are very exciting and I view them as being a major contribution. Games such as Dark Chess/Recon Chess/Stratego have recently emerged as the next benchmark for the 2p zero-sum research community, and this paper is the first to achieve very strong performance on these types of very large games. However, I would like to see discussion in the paper of how general this method is. Section 5 seems to include a lot of domain-specific tricks, which makes me less sure that it can generalize to a game like Stratego. Furthermore, I am not sure why this method works on Dark Chess. A possible explanation the authors give is that |I^1| is much less than |I^{\infty}| in Dark Chess but not in poker. Do the authors have any evidence for this claim? “In such settings [poker], we do not expect our techniques to give much improvement over the current state of the art” should include clearer explanation for why: is it that “I^3 = I^∞ for every I (and I^2 is already very close to I^∞, excluding only “blockers”)“, or is it “Further, I^∞ itself is quite small”? Perhaps the information of the sizes of |I^k| for different k can be included in the experiments. I don’t see how Dark Chess has a fundamentally different information structure from poker, in that the private information in Dark Chess can be viewed as a hand in poker, so some further explanation of this would be great. For example, would this method work for Gin Rummy? Also, isn’t |I^1| still prohibitively large in Dark Chess? How are these subgames solved tractably?

I am confused about how you can throw out the nodes in $\bar{I^{k+1}} \setminus \bar{I^k}$. In section 4 item 2 the authors say that “the payoff at these nodes is only a function of ⊖’s strategy in the subgame and the blueprint strategy.” but this doesn’t make sense to me. When running CFR on the gadget game, ⊖’s policy in the subgame changes every iteration, so the payoff for each node of that kind needs to be recomputed every iteration. Is just this a simply a simplification? Should these nodes actually be recomputed but instead it works empirically if we replace them with static values from the blueprint?

Additionally, I would like to see more details, maybe in the appendix, about how the Dark Chess AI was created. For example, how was the subgame solved (CFR, LP?), what are all the hyperparameters?

Overall, I am excited by the results in this paper and by the novel method, so I would like to see the paper accepted, but I think the paper could be made stronger with the suggestions above.

__________________
I have read the rebuttal and do not change my score. Thank you for the detailed responses to my questions, that clears a lot up.

**Time Spent Reviewing:**

8

---

> ### Author Response · Authors · 2021-08-11
> **Respose to Reviewer MEQH**
>
> Thank you for the review!
>
> **My main concern with this paper is that the technique is not safe. Although the authors provide theorems 4 and 5, I don’t think these theorems will change the method to make it safe. Theorem 4 is technically correct, but I don’t think it amounts to anything that will make the technique actually safe. Going back to the counterexample in proposition 3, if we play this game against 1-KLSS, we can always exploit it because we know it will always play tails. Theorem 4 doesn’t solve this problem because it only updates the blueprint after you solve the subgame and play the action. Theorem 5 also seems weak to me because it starts with the assumption that you already have an epsilon-Nash, which is a strong assumption and in which case you could just do nothing and keep an epsilon-Nash. While affine equilibrium could be a decent solution concept (I’m not quite sure), theorem 9 also assumes that you start with a Nash equilibrium blueprint. Since there are also no experiments that use or validate these theorems, I am unconvinced that they can be used to make the technique safe in practice. With that being said, I give credit to the authors for pointing out the limitations of the method, for example in the last two paragraphs of section 4.**
>
> These comments are fair, and we will clarify them further in the camera-ready version.
>
> As you state, the theorems are correct. To recover a full safety guarantee from Thm 4, the blueprint--not the result of the subgame solve--should be used during play, and the only function of the subgame solve is to update the blueprint for later use. Similarly, to recover a full safety guarantee for Thm 5, the collection of information sets at which to do subgame solving (or a probability distribution over such collections) should be fixed in advance. We will state this clearly.
>
> The assumption of starting from epsilon-equilibrium in Thm 5 is standard when discussing safety in the subgame solving literature, where safety means that exploitability cannot increase. We acknowledge that Thm 9 is rather weak (indeed, if you already knew you had a Nash blueprint, you wouldn't run subgame solving), but the theorem statement serves as an illustration of (roughly) what our method achieves in lieu of not having the type of safety guarantee that prior safe subgame solving techniques (which are unscalable for this problem) have.
>
> In any case, the version of subgame solving used in the experiments, both in dark chess and the medium-sized games, does not use any of the mitigation techniques, so the experiments are unaffected by this discussion, and demonstrate safety in practice despite a lack of theoretical guarantees.
>
> **For example, in the medium-size game experiments, I am not convinced that adding epsilon-uniform noise is enough to systematically bias the blueprint. Instead, if you added epsilon-fold noise (where it folds with epsilon probability) for poker games, my guess would be that the new strategy would be worse than the old strategy.**
>
> Good question. As a response to your question, we ran a quick additional experiment. The result suggests that your guess is incorrect. In the poker games that we used in the experiments, if 0.25-fold probability is added to the blueprint, the strategy after subgame solving has the same exploitability as the strategy before subgame solving. If we increase epsilon, the strategy after subgame solving once again has lower exploitability than the strategy before. We will include this in the camera-ready version of the paper.
>
> **I like that many medium-sized games were used, but the results give the impression that 1-KLSS will always be less exploitable than the blueprint, which is not the case for different blueprints. In general, I would like to see more theoretical/empirical exploration of the types of games/blueprints where 1-KLSS performs well and where it will fail (such as the counterexample).**
>
> We will include the above quick experiment in the camera-ready version. If there are additional experiments the reviewer wants to suggest, we would be happy to consider them.
>
> **I am not sure why this method works on Dark Chess. A possible explanation the authors give is that |I^1| is much less than |I^∞| in Dark Chess but not in poker. Do the authors have any evidence for this claim?**
>
> In poker, |I^1| is the number of possible hole-card combos for one player, which is 1326. |I^∞| is the number of hole-card combos for both players, which is about 1.6 million--that is a reasonably small enough number to work with in real time. In dark chess, |I^1| is at most, say, 10^7 in our experience (which again is small enough to work with in real time), while any attempt to enumerate I^∞ resulted in very fast out-of-memory errors.
>
> **“In such settings [poker], we do not expect our techniques to give much improvement over the current state of the art” should include clearer explanation for why: is it that “I^3 = I^∞ for every I (and I^2 is already very close to I^∞, excluding only “blockers”)“, or is it “Further, I^∞ itself is quite small”?**
>
> The latter. The whole point of KLSS is to have something to use in cases where common-knowledge subgame solving is not computationally feasible. In poker (at least, Texas hold’em), I^∞ is quite small, so common-knowledge subgame solving is already tractable--so we should just use that instead, since it has stronger safety guarantees.
>
> The comment about I^3 = I^∞ was more to draw a sharp contrast to the k-MP game in the previous paragraph. We will clarify this in the camera-ready version.
>
> **Perhaps the information of the sizes of |I^k| for different k can be included in the experiments.**
>
> We will include this in the camera-ready version. It is worth mentioning that the medium-sized games in the experiments are all very “poker-like” more than “dark chess-like”, in that I^∞ is small. The intended purpose of experiments in these smaller games was to demonstrate that KLSS, even without any mitigation method, is still usually safe in practice--not to compare performance with safer methods such as common-knowledge subgame solving, which do not scale to large games.
>
> **I don’t see how Dark Chess has a fundamentally different information structure from poker, in that the private information in Dark Chess can be viewed as a hand in poker, so some further explanation of this would be great.**
>
> Here is a simple example of how high-order non-common knowledge can arise in dark chess. After the moves 1. a3 e5 2. Nf3 Qf6, the statement P = "there is a pawn on a3" is fourth-order knowledge (Black knows that White knows that Black knows that White knows P), but not higher-order (White doesn't know that Black knows that White knows that Black knows that White knows P).
>
> Poker cannot have such statements--the highest-order uncommon knowledge that arises in poker is a blocker: if I have a hole card C, then the statement P = "You do not have the hole card C" is second-order (I know that you know P) but not common knowledge (you do not know that I know that you know P).
>
> **For example, would this method work for Gin Rummy?**
>
> No, |I^1| is too big in Gin Rummy (having size 42 choose 10 ~= 10^12 at the beginning of the game, and possibly increasing later). Further developments would be needed. Same goes for Stratego, where |I^1| has absurd size from the first move.
>
> **Also, isn’t |I^1| still prohibitively large in Dark Chess? How are these subgames solved tractably?**
>
> |I^1| is about 10^7 at most in dark chess (and usually smaller). This is feasible to enumerate in real time. We also use some practical heuristics to subsample from |I^1| so that we can search to higher depths. We discuss both of these points in the paper (see "Dealing with lost particles", line 343).
>
> **I am confused about how you can throw out the nodes in. In section 4 item 2 the authors say that “the payoff at these nodes is only a function of ⊖’s strategy in the subgame and the blueprint strategy.” but this doesn’t make sense to me. When running CFR on the gadget game, ⊖’s policy in the subgame changes every iteration, so the payoff for each node of that kind needs to be recomputed every iteration. Is just this a simply a simplification? Should these nodes actually be recomputed but instead it works empirically if we replace them with static values from the blueprint?**
>
> Yes, ⊖'s subgame policy changes in the subgame solve, but that's fine--all we are doing is observing that there is no dependence on ⊕'s strategy (since it is fixed)--and so, in the subgame, the terminal nodes in these sections of the tree end up in ⊕'s root sequence in the payoff matrix. This is more of an observation of what happens during the subgame solve than an actual simplification--the nodes are still there, it's just that they don't actually appear except in a single row of the payoff matrix, reducing the complexity. See the example in the appendix, in particular the construction of matrix B^{Γ[R_1]} in lines 674-678, for details.
>
> **Additionally, I would like to see more details, maybe in the appendix, about how the Dark Chess AI was created. For example, how was the subgame solved (CFR, LP?), what are all the hyperparameters?**
>
> Subgames are solved exactly via LP, using Gurobi on default settings. All relevant hyperparameters are stated.

---

### Official Review · Reviewer_Dn9V · 2021-07-16

**Rating:** 6
**Confidence:** 4

**Summary:**

This work proposes a new method for subgame solving in two-player zero-sum extensive-form games with imperfect information. Traditional subgame solvers rely on the so-called common-knowledge closure that guarantees the opponent can not exploit the strategy computed in the subgame in the whole game. The problem with the common-knowledge closure is that its size may be beyond the abilities of contemporary solvers in case the game does not reveal enough information observable by both players during the course of play. To enable solving even subgames with large common-knowledge closures of the information set in the root, the authors introduce order-k knowledge sets as sets of nodes in the game tree that are reachable from the base set in at most k-1hyperedges in the infoset hypergraph. The order-k knowledge sets reach the common-knowledge closure as k approaches infinity. The order-k knowledge sets may be much smaller than the common-knowledge sets, which results in more efficient subgame solving, but this comes at the expense of the exploitability of the computed strategy. As the authors show, the exploitability may increase by a factor linear in the size of the game. The authors hence propose two methods for mitigating it. The first method requires updating the blueprint strategy after every iteration of subgame solving, while the second method restricts the possible deviations from the blueprint strategy. Both methods guarantee that the exploitability of the new strategy will not exceed the exploitability of the blueprint. Using the order-1 knowledge-limited subgame solver as well as several other heuristics, the authors built an agent for playing a large extensive-form game called dark chess. In the experimental part of the manuscript, the authors first show that their method successfully decreases the exploitability of the blueprint strategy for seven medium-sized games. In the second part, they report results achieved with the agent playing dark chess. The agent can beat a baseline player and an amateur human player but loses against the current champion.


**Ethical Concerns:**

I do not have any ethical concerns.

**Limitations And Societal Impact:**

The authors mention the limitations of their technique and describe settings when the technique may not provide significant improvements over the common-knowledge closure, which I appreciate. I still believe some experimental comparison with the standard subgame solvers is desirable, though.

**Main Review:**

The idea of a limited-knowledge closure as a restriction of the common-knowledge closure is indeed natural and interesting enough to warrant further investigation. The theoretical results presented by the authors are relatively simple. Still, the proofs seem correct, and the methods proposed in Theorems 4 and 5 mitigate the main drawbacks of subgame solving based on order-k knowledge sets. I am a bit confused regarding motivation for Theorem 9, though. I understand it shows that 1-KLSS reaches a strategy that is the best response but may not be equilibrial, thus in a way mitigating exploitability even without employing methods introduced earlier. However, why would we need to solve a subgame if we already have a Nash blueprint strategy at hand? What is the point?

Moreover, I believe the experimental part does not present enough evidence to support the authors’ claims fully. The primary motivation of the paper is to solve games where common-knowledge closure is too large for standard subgame solvers to compute a solution. Therefore, I would expect to find some experimental comparison with these subgame solvers in this work in terms of runtimes, memory requirements, and decreases in exploitability. For completeness, it would also help to show how big the games are (number of information sets, number of sequences, etc.) and what the difference is in sizes of common-knowledge closures and order-1 knowledge sets. If I understand it correctly, the version of 1-KLSS used in the experiments does not employ the exploitability mitigation methods. How does the exploitability (and also runtime, memory requirements) change when they are used? Do the authors have some intuition why 1-KLSS manages to refine the strategy well in some cases, but in others (e.g., dark hex or Leduc poker), it remains mostly unchanged?

Otherwise, I find the text of the manuscript to be fairly well written. The example in Appendix D illustrates well the differences between using the common-knowledge closures and order-1 knowledge sets. I would argue that including at least a shortened version of the example in the main text may significantly help the readers to familiarize themselves with the introduced concepts and may serve a better purpose than, e.g., the description of tricks used in the dark chess player (even though its performance is an impressive feat). The language of some of the definitions and claims also feels a bit too colloquial and vague at times, especially later in the paper, e.g., in Proposition 8, Theorem 9, and Definition 10.

Few other nitpicks:
Line 45: I do not think it is fair to call order-k knowledge sets a “completely different technique” when they in fact generalize the common-knowledge closures.
“Kth-order knowledge” is mentioned in the introduction (line 46) without any (even informal) explanation.
It may also help to (at least informally) explain what a “blueprint” strategy is and what purpose it serves, as it may not be a common-knowledge among all the readers.


### After rebuttal

Thank you for your response.


**Time Spent Reviewing:**

25

---

> ### Author Response · Authors · 2021-08-11
> **Respose to Reviewer Dn9V**
>
> Thank you for the review!
>
> **The idea of a limited-knowledge closure as a restriction of the common-knowledge closure is indeed natural and interesting enough to warrant further investigation. The theoretical results presented by the authors are relatively simple. Still, the proofs seem correct, and the methods proposed in Theorems 4 and 5 mitigate the main drawbacks of subgame solving based on order-k knowledge sets. I am a bit confused regarding motivation for Theorem 9, though. I understand it shows that 1-KLSS reaches a strategy that is the best response but may not be equilibrial, thus in a way mitigating exploitability even without employing methods introduced earlier. However, why would we need to solve a subgame if we already have a Nash blueprint strategy at hand? What is the point?**
>
> Yes, we acknowledge that Thm 9 is rather weak (indeed, if you already knew you had a Nash blueprint, you wouldn't run subgame solving), but the theorem statement serves as an illustration of (roughly) what our method achieves in lieu of not having the type of safety guarantee that prior safe subgame solving techniques (which are unscalable for this problem) have.
>
> **Moreover, I believe the experimental part does not present enough evidence to support the authors’ claims fully. The primary motivation of the paper is to solve games where common-knowledge closure is too large for standard subgame solvers to compute a solution. Therefore, I would expect to find some experimental comparison with these subgame solvers in this work in terms of runtimes, memory requirements, and decreases in exploitability. For completeness, it would also help to show how big the games are (number of information sets, number of sequences, etc.) and what the difference is in sizes of common-knowledge closures and order-1 knowledge sets. If I understand it correctly, the version of 1-KLSS used in the experiments does not employ the exploitability mitigation methods. How does the exploitability (and also runtime, memory requirements) change when they are used?**
>
> We will include the game statistics and some more detailed experiments in the camera-ready version.
>
> It is worth mentioning that the intended purpose of experiments in these smaller games was to demonstrate that KLSS, even without any mitigation method, is still usually safe in practice--not to compare performance with safer methods such as common-knowledge subgame solving. As we mention multiple times in the paper, we do not recommend using KLSS when the game is simple enough that common-knowledge subgame solving is feasible. The key purpose of this paper is to introduce a technique that works when it is not.
>
> **Do the authors have some intuition why 1-KLSS manages to refine the strategy well in some cases, but in others (e.g., dark hex or Leduc poker), it remains mostly unchanged?**
>
> Good question. Unfortunately, not really.
>
> **I would argue that including at least a shortened version of the example in the main text may significantly help the readers to familiarize themselves with the introduced concepts and may serve a better purpose than, e.g., the description of tricks used in the dark chess player (even though its performance is an impressive feat). The language of some of the definitions and claims also feels a bit too colloquial and vague at times, especially later in the paper, e.g., in Proposition 8, Theorem 9, and Definition 10.**
>
> In the camera-ready version, we will make this swap.
>
> **Few other nitpicks: Line 45: I do not think it is fair to call order-k knowledge sets a “completely different technique” when they in fact generalize the common-knowledge closures. “Kth-order knowledge” is mentioned in the introduction (line 46) without any (even informal) explanation. It may also help to (at least informally) explain what a “blueprint” strategy is and what purpose it serves, as it may not be a common-knowledge among all the readers.**
>
> We will make these changes in the camera-ready.

---

### Official Review · Reviewer_kM15 · 2021-07-19

**Rating:** 7
**Confidence:** 3

**Summary:**

The authors develop a new approximation to for subgame solving that approximate common knowledge using lower-order knowledge. Using theoretical and empirical tests they show that these algorithms can reduce exploitability when the information sets are large. Finally, they use these ideas to develop a game playing agent for dark chess and test it against algorithms, a coauthor and the current top online player.

**Limitations And Societal Impact:**

See above.

**Main Review:**

Originality: To my knowledge this work is original and most excitingly this is one of the first game playing agents for dark chess. The research direction and prior art are clearly stated and the problem addressed in this work is both timely and at the frontier of what could be possible.

Line 27: "massive challenge problem" is unnecessary hype
Line 60-61: "massive benchmark game" again is hype -- there is no citation for this game.r chess).

Quality: I would have liked to see empirical evaluations for higher k for at least some of the games even if there aren't theoretical guarantees available. I would also have liked to see exploitability improvements (or possible failures) from a more diverse set of blueprint strategies than just \epsilon-uniform.


The results on dark chess are quite exciting but the best evidence that the agent is playing at the level of an amateur human comes from playing against a co-author. The possible bias of this kind of evaluation (there is a strong (unconscious) incentive to lose to get the paper into NeurIPS!) should at least be mentioned and giving the agent an ELO rating based on these two data points feels like building on top of what is already a relatively small sample size.

I did not review the proofs for correctness.

Clarity: The paper could be organized better. In particular I would like to see a more succinct outline of the dark chess game playing agent. I would like to see a more clear contrast of what we have learned from imperfect information zero-sum games from poker and what we will learn from dark chess as they push on different aspects of the problem. Perhaps most interestingly would be a direction where a single algorithm could yield superhuman performance in both settings and this possibility could be commented on.

Is this the first work to develop a game playing agent for this game? Are there other baselines available besides a lesion of this agent? Is the Stockfish agent being run with maximum settings on the comparable hardware?

Significance: This paper passes the bar as a significant contribution with theoretical results, empirical validation and quantification and an exciting man vs. machine bake-off. I hope that by the time of the conference the experiments can be improved. Having the agent play more humans would significantly strengthen the claims here.

**Time Spent Reviewing:**

3

---

> ### Author Response · Authors · 2021-08-11
> **Response to Reviewer kM15**
>
> Thank you for the review!
>
> **Line 27: "massive challenge problem" is unnecessary hype Line 60-61: "massive benchmark game" again is hype -- there is no citation for this game.r chess).**
>
> Fair--"massive" was intended to refer more to the size/difficulty of the game. We will reword for clarity in the camera-ready.
>
> **I would have liked to see empirical evaluations for higher k for at least some of the games even if there aren't theoretical guarantees available. I would also have liked to see exploitability improvements (or possible failures) from a more diverse set of blueprint strategies than just \epsilon-uniform.**
>
> We will include some experiments for higher k and different blueprints in the camera-ready version. Here is one such result: as we mention in another response, if we start with blueprint that has forced 0.25-fold probability in the poker variants, running 1-KLSS does not increase nor decrease exploitability. If we increase epsilon, the strategy after subgame solving once again has lower exploitability than the strategy before.
>
> **The results on dark chess are quite exciting but the best evidence that the agent is playing at the level of an amateur human comes from playing against a co-author. The possible bias of this kind of evaluation (there is a strong (unconscious) incentive to lose to get the paper into NeurIPS!) should at least be mentioned and giving the agent an ELO rating based on these two data points feels like building on top of what is already a relatively small sample size.**
>
> Fair--of course, I (the author who played the agent) did not intentionally weaken my play, but the concern is reasonable. Perhaps the strongest other evidence is the actual games played against the world’s best human player--despite the fact that our agent only won one of the ten games, the fact that it (1) still won one, and (2) exhibited (in our view) strong play in the other games as well -- especially in the early and mid-games where it seemed to often outperform the world’s best human -- demonstrates strong performance.
>
> **The paper could be organized better. In particular I would like to see a more succinct outline of the dark chess game playing agent.**
>
> Another reviewer suggested moving the full description of the dark chess agent to the appendix, and leaving only an outline in the body, which we will do.
>
> **I would like to see a more clear contrast of what we have learned from imperfect information zero-sum games from poker and what we will learn from dark chess as they push on different aspects of the problem.**
>
> As we emphasize throughout the paper, the main contrast is that poker has small common-knowledge sets, and dark chess does not. Thus, traditional subgame solving techniques based on common-knowledge subgame solving cannot be applied in dark chess, which is the motivation for this paper.
>
> **Is this the first work to develop a game playing agent for this game? Are there other baselines available besides a lesion of this agent?**
>
> To our knowledge, this is the first AI on this game.
>
> **Is the Stockfish agent being run with maximum settings on the comparable hardware?**
>
> Yes.
>
> **This paper passes the bar as a significant contribution with theoretical results, empirical validation and quantification and an exciting man vs. machine bake-off. I hope that by the time of the conference the experiments can be improved. Having the agent play more humans would significantly strengthen the claims here.**
>
> Playing more humans is a challenge here due to logistical constraints: chess dot com does not allow bots to run in their open lobbies, so we must set up individual matches. Instead of spending a huge amount of effort to evaluate exactly where our bot would fare in a field of non-top humans, our goal going forward is to somehow improve the bot and then go against the best player in the world again, hoping that we can achieve superhuman performance.

---

> > ### Comment · Reviewer_kM15 · 2021-08-22
> > **Thank you**
> >
> > Thank you for the detailed response. I support this paper be accepted

---

### Decision · Program_Chairs · 2021-09-27

**Decision:**

Accept (Spotlight)

**Comment:**

This appears to be a solid paper with strong reviews. The reviewer concerns were addressed with clarity and detail by the authors. I hope that any final version of the paper will incorporate improvements as a result of the reviewer feedback.